# How Powerful are Graph Neural Networks?

**Keyulu Xu** [*][†]
MIT
keyulu@mit.edu

**Weihua Hu** [*][‡]
Stanford University
weihuahu@stanford.edu

**Jure Leskovec**
Stanford University
jure@cs.stanford.edu

**Stefanie Jegelka**
MIT
stefje@mit.edu

## Abstract

Graph Neural Networks (GNNs) are an effective framework for representation learning of graphs. GNNs follow a neighborhood aggregation scheme, where the representation vector of a node is computed by recursively aggregating and transforming representation vectors of its neighboring nodes. Many GNN variants have been proposed and have achieved state-of-the-art results on both node and graph classification tasks. However, despite GNNs revolutionizing graph representation learning, there is limited understanding of their representational properties and limitations. Here, we present a theoretical framework for analyzing the expressive power of GNNs to capture different graph structures. Our results characterize the discriminative power of popular GNN variants, such as Graph Convolutional Networks and GraphSAGE, and show that they cannot learn to distinguish certain simple graph structures. We then develop a simple architecture that is provably the most expressive among the class of GNNs and is as powerful as the Weisfeiler-Lehman graph isomorphism test. We empirically validate our theoretical findings on a number of graph classification benchmarks, and demonstrate that our model achieves state-of-the-art performance.

## 1 Introduction

Learning with graph structured data, such as molecules, social, biological, and financial networks, requires effective representation of their graph structure (Hamilton et al., 2017b). Recently, there has been a surge of interest in Graph Neural Network (GNN) approaches for representation learning of graphs (Li et al., 2016; Hamilton et al., 2017a; Kipf & Welling, 2017; Velickovic et al., 2018; Xu et al., 2018). GNNs broadly follow a recursive neighborhood aggregation (or message passing) scheme, where each node aggregates feature vectors of its neighbors to compute its new feature vector (Xu et al., 2018; Gilmer et al., 2017). After $k$ iterations of aggregation, a node is represented by its transformed feature vector, which captures the structural information within the node's $k$-hop neighborhood. The representation of an entire graph can then be obtained through pooling (Ying et al., 2018), for example, by summing the representation vectors of all nodes in the graph.

Many GNN variants with different neighborhood aggregation and graph-level pooling schemes have been proposed (Scarselli et al., 2009b; Battaglia et al., 2016; Defferrard et al., 2016; Duvenaud et al., 2015; Hamilton et al., 2017a; Kearnes et al., 2016; Kipf & Welling, 2017; Li et al., 2016; Velickovic et al., 2018; Santoro et al., 2017; Xu et al., 2018; Santoro et al., 2018; Verma & Zhang, 2018; Ying et al., 2018; Zhang et al., 2018). Empirically, these GNNs have achieved state-of-the-art performance in many tasks such as node classification, link prediction, and graph classification. However, the design of new GNNs is mostly based on empirical intuition, heuristics, and experimental trial-and-error. There is little theoretical understanding of the properties and limitations of GNNs, and formal analysis of GNNs' representational capacity is limited.

---

[*]Equal contribution.
[†]Work partially performed while in Tokyo, visiting Prof. Ken-ichi Kawarabayashi.
[‡]Work partially performed while at RIKEN AIP and University of Tokyo.

Here, we present a theoretical framework for analyzing the representational power of GNNs. We formally characterize how expressive different GNN variants are in learning to represent and distinguish between different graph structures. Our framework is inspired by the close connection between GNNs and the Weisfeiler-Lehman (WL) graph isomorphism test (Weisfeiler & Lehman, 1968), a powerful test known to distinguish a broad class of graphs (Babai & Kucera, 1979). Similar to GNNs, the WL test iteratively updates a given node's feature vector by aggregating feature vectors of its network neighbors. What makes the WL test so powerful is its injective aggregation update that maps different node neighborhoods to different feature vectors. Our key insight is that a GNN can have as large discriminative power as the WL test if the GNN's aggregation scheme is highly expressive and can model injective functions.

To mathematically formalize the above insight, our framework first represents the set of feature vectors of a given node's neighbors as a *multiset*, *i.e.*, a set with possibly repeating elements. Then, the neighbor aggregation in GNNs can be thought of as an *aggregation function over the multiset*. Hence, to have strong representational power, a GNN must be able to aggregate different multisets into different representations. We rigorously study several variants of multiset functions and theoretically characterize their discriminative power, *i.e.*, how well different aggregation functions can distinguish different multisets. The more discriminative the multiset function is, the more powerful the representational power of the underlying GNN.

Our main results are summarized as follows:

1) We show that GNNs are *at most* as powerful as the WL test in distinguishing graph structures.

2) We establish conditions on the neighbor aggregation and graph readout functions under which the resulting GNN is *as powerful as* the WL test.

3) We identify graph structures that cannot be distinguished by popular GNN variants, such as GCN (Kipf & Welling, 2017) and GraphSAGE (Hamilton et al., 2017a), and we precisely characterize the kinds of graph structures such GNN-based models can capture.

4) We develop a simple neural architecture, *Graph Isomorphism Network (GIN)*, and show that its discriminative/representational power is equal to the power of the WL test.

We validate our theory via experiments on graph classification datasets, where the expressive power of GNNs is crucial to capture graph structures. In particular, we compare the performance of GNNs with various aggregation functions. Our results confirm that the most powerful GNN by our theory, *i.e.*, Graph Isomorphism Network (GIN), also empirically has high representational power as it almost perfectly fits the training data, whereas the less powerful GNN variants often severely underfit the training data. In addition, the representationally more powerful GNNs outperform the others by test set accuracy and achieve state-of-the-art performance on many graph classification benchmarks.

## 2 PRELIMINARIES

We begin by summarizing some of the most common GNN models and, along the way, introduce our notation. Let $G = (V, E)$ denote a graph with node feature vectors $X_v$ for $v \in V$. There are two tasks of interest: (1) *Node classification*, where each node $v \in V$ has an associated label $y_v$ and the goal is to learn a representation vector $h_v$ of $v$ such that $v$'s label can be predicted as $y_v = f(h_v)$; (2) *Graph classification*, where, given a set of graphs $\{G_1, ..., G_N\} \subseteq \mathcal{G}$ and their labels $\{y_1, ..., y_N\} \subseteq \mathcal{Y}$, we aim to learn a representation vector $h_G$ that helps predict the label of an entire graph, $y_G = g(h_G)$.

**Graph Neural Networks.** GNNs use the graph structure and node features $X_v$ to learn a representation vector of a node, $h_v$, or the entire graph, $h_G$. Modern GNNs follow a neighborhood aggregation strategy, where we iteratively update the representation of a node by aggregating representations of its neighbors. After $k$ iterations of aggregation, a node's representation captures the structural information within its $k$-hop network neighborhood. Formally, the $k$-th layer of a GNN is

$$a_v^{(k)} = \text{AGGREGATE}^{(k)} \left( \left\{ h_u^{(k-1)} : u \in \mathcal{N}(v) \right\} \right), \quad h_v^{(k)} = \text{COMBINE}^{(k)} \left( h_v^{(k-1)}, a_v^{(k)} \right),$$
(2.1)

where $h_v^{(k)}$ is the feature vector of node $v$ at the $k$-th iteration/layer. We initialize $h_v^{(0)} = X_v$, and $\mathcal{N}(v)$ is a set of nodes adjacent to $v$. The choice of $\text{AGGREGATE}^{(k)}(\cdot)$ and $\text{COMBINE}^{(k)}(\cdot)$ in

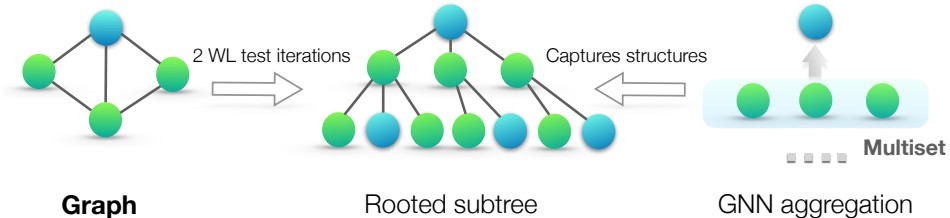

Figure 1: **An overview of our theoretical framework.** Middle panel: rooted subtree structures (at the blue node) that the WL test uses to distinguish different graphs. Right panel: if a GNN's aggregation function captures the *full multiset* of node neighbors, the GNN can capture the rooted subtrees in a recursive manner and be as powerful as the WL test.

GNNs is crucial. A number of architectures for AGGREGATE have been proposed. In the pooling variant of GraphSAGE (Hamilton et al., 2017a), AGGREGATE has been formulated as

$$a_v^{(k)} = \text{MAX}\left(\left\{\text{ReLU}\left(W \cdot h_u^{(k-1)}\right), \forall u \in \mathcal{N}(v)\right\}\right), \tag{2.2}$$

where $W$ is a learnable matrix, and MAX represents an element-wise max-pooling. The COMBINE step could be a concatenation followed by a linear mapping $W \cdot \left[h_v^{(k-1)}, a_v^{(k)}\right]$ as in GraphSAGE. In Graph Convolutional Networks (GCN) (Kipf & Welling, 2017), the element-wise *mean* pooling is used instead, and the AGGREGATE and COMBINE steps are integrated as follows:

$$h_v^{(k)} = \text{ReLU}\left(W \cdot \text{MEAN}\left\{h_u^{(k-1)}, \forall u \in \mathcal{N}(v) \cup \{v\}\right\}\right). \tag{2.3}$$

Many other GNNs can be represented similarly to Eq. 2.1 (Xu et al., 2018; Gilmer et al., 2017).

For node classification, the node representation $h_v^{(K)}$ of the final iteration is used for prediction. For graph classification, the READOUT function aggregates node features from the final iteration to obtain the entire graph's representation $h_G$:

$$h_G = \text{READOUT}\left(\left\{h_v^{(K)} \mid v \in G\right\}\right). \tag{2.4}$$

READOUT can be a simple permutation invariant function such as summation or a more sophisticated graph-level pooling function (Ying et al., 2018; Zhang et al., 2018).

**Weisfeiler-Lehman test.** The graph isomorphism problem asks whether two graphs are topologically identical. This is a challenging problem: no polynomial-time algorithm is known for it yet (Garey, 1979; Garey & Johnson, 2002; Babai, 2016). Apart from some corner cases (Cai et al., 1992), the Weisfeiler-Lehman (WL) test of graph isomorphism (Weisfeiler & Lehman, 1968) is an effective and computationally efficient test that distinguishes a broad class of graphs (Babai & Kucera, 1979). Its 1-dimensional form, "naïve vertex refinement", is analogous to neighbor aggregation in GNNs. The WL test iteratively (1) aggregates the labels of nodes and their neighborhoods, and (2) hashes the aggregated labels into *unique* new labels. The algorithm decides that two graphs are non-isomorphic if at some iteration the labels of the nodes between the two graphs differ.

Based on the WL test, Shervashidze et al. (2011) proposed the WL subtree kernel that measures the similarity between graphs. The kernel uses the counts of node labels at different iterations of the WL test as the feature vector of a graph. Intuitively, a node's label at the $k$-th iteration of WL test represents a subtree structure of height $k$ rooted at the node (Figure 1). Thus, the graph features considered by the WL subtree kernel are essentially counts of different rooted subtrees in the graph.

## 3 THEORETICAL FRAMEWORK: OVERVIEW

We start with an overview of our framework for analyzing the expressive power of GNNs. Figure 1 illustrates our idea. A GNN recursively updates each node's feature vector to capture the network structure and features of other nodes around it, *i.e.*, its rooted subtree structures (Figure 1). Throughout the paper, we assume node input features are from a countable universe. For finite graphs, node feature vectors at deeper layers of any fixed model are also from a countable universe. For notational simplicity, we can assign each feature vector a unique label in $\{a, b, c \ldots\}$. Then, feature vectors of a set of neighboring nodes form a *multiset* (Figure 1): the same element can appear multiple times since different nodes can have identical feature vectors.

**Definition 1** (Multiset). *A multiset is a generalized concept of a set that allows multiple instances for its elements. More formally, a multiset is a 2-tuple $X = (S, m)$ where $S$ is the* underlying set *of $X$ that is formed from its* distinct elements, *and $m : S \to \mathbb{N}_{\geq 1}$ gives the* multiplicity *of the elements.*

To study the representational power of a GNN, we analyze when a GNN maps two nodes to the same location in the embedding space. Intuitively, a maximally powerful GNN maps two nodes to the same location *only if* they have identical subtree structures with identical features on the corresponding nodes. Since subtree structures are defined recursively via node neighborhoods (Figure 1), we can reduce our analysis to the question whether a GNN maps two neighborhoods (*i.e.*, two multisets) to the same embedding or representation. A maximally powerful GNN would *never* map two different neighborhoods, *i.e.*, multisets of feature vectors, to the same representation. This means its aggregation scheme must be *injective*. Thus, we abstract a GNN's aggregation scheme as a class of functions over multisets that their neural networks can represent, and analyze whether they are able to represent injective multiset functions.

Next, we use this reasoning to develop a maximally powerful GNN. In Section 5, we study popular GNN variants and see that their aggregation schemes are inherently not injective and thus less powerful, but that they can capture other interesting properties of graphs.

## 4 BUILDING POWERFUL GRAPH NEURAL NETWORKS

First, we characterize the maximum representational capacity of a general class of GNN-based models. Ideally, a maximally powerful GNN could distinguish different graph structures by mapping them to different representations in the embedding space. This ability to map any two different graphs to different embeddings, however, implies solving the challenging graph isomorphism problem. That is, we want isomorphic graphs to be mapped to the same representation and non-isomorphic ones to different representations. In our analysis, we characterize the representational capacity of GNNs via a slightly weaker criterion: a powerful heuristic called *Weisfeiler-Lehman (WL) graph isomorphism test*, that is known to work well in general, with a few exceptions, e.g., regular graphs (Cai et al., 1992; Douglas, 2011; Evdokimov & Ponomarenko, 1999).

**Lemma 2.** *Let $G_1$ and $G_2$ be any two non-isomorphic graphs. If a graph neural network $\mathcal{A} : \mathcal{G} \to \mathbb{R}^d$ maps $G_1$ and $G_2$ to different embeddings, the Weisfeiler-Lehman graph isomorphism test also decides $G_1$ and $G_2$ are not isomorphic.*

Proofs of all Lemmas and Theorems can be found in the Appendix. Hence, any aggregation-based GNN is at most as powerful as the WL test in distinguishing different graphs. A natural follow-up question is whether there exist GNNs that are, in principle, as powerful as the WL test? Our answer, in Theorem 3, is yes: if the neighbor aggregation and graph-level readout functions are injective, then the resulting GNN is as powerful as the WL test.

**Theorem 3.** *Let $\mathcal{A} : \mathcal{G} \to \mathbb{R}^d$ be a GNN. With a sufficient number of GNN layers, $\mathcal{A}$ maps any graphs $G_1$ and $G_2$ that the Weisfeiler-Lehman test of isomorphism decides as non-isomorphic, to different embeddings if the following conditions hold:*

*a) $\mathcal{A}$ aggregates and updates node features iteratively with*

$$h_v^{(k)} = \phi\left(h_v^{(k-1)}, f\left(\left\{h_u^{(k-1)} : u \in \mathcal{N}(v)\right\}\right)\right),$$

*where the functions $f$, which operates on multisets, and $\phi$ are injective.*

*b) $\mathcal{A}$'s graph-level readout, which operates on the multiset of node features $\left\{h_v^{(k)}\right\}$, is injective.*

We prove Theorem 3 in the appendix. For countable sets, injectiveness well characterizes whether a function preserves the distinctness of inputs. Uncountable sets, where node features are continuous, need some further considerations. In addition, it would be interesting to characterize how close together the learned features lie in a function's image. We leave these questions for future work, and focus on the case where input node features are from a countable set (that can be a subset of an uncountable set such as $\mathbb{R}^n$).

**Lemma 4.** *Assume the input feature space $\mathcal{X}$ is countable. Let $g^{(k)}$ be the function parameterized by a GNN's $k$-th layer for $k = 1, ..., L$, where $g^{(1)}$ is defined on multisets $X \subset \mathcal{X}$ of bounded size. The range of $g^{(k)}$, i.e., the space of node hidden features $h_v^{(k)}$, is also countable for all $k = 1, ..., L$.*

Here, it is also worth discussing an important benefit of GNNs beyond distinguishing different graphs, that is, capturing similarity of graph structures. Note that node feature vectors in the WL test are essentially one-hot encodings and thus cannot capture the similarity between subtrees. In contrast, a GNN satisfying the criteria in Theorem 3 generalizes the WL test by *learning to embed* the subtrees to low-dimensional space. This enables GNNs to not only discriminate different structures, but also to learn to map similar graph structures to similar embeddings and capture dependencies between graph structures. Capturing structural similarity of the node labels is shown to be helpful for generalization particularly when the co-occurrence of subtrees is sparse across different graphs or there are noisy edges and node features (Yanardag & Vishwanathan, 2015).

## 4.1 GRAPH ISOMORPHISM NETWORK (GIN)

Having developed conditions for a maximally powerful GNN, we next develop a simple architecture, *Graph Isomorphism Network (GIN)*, that provably satisfies the conditions in Theorem 3. This model generalizes the WL test and hence achieves maximum discriminative power among GNNs.

To model injective multiset functions for the neighbor aggregation, we develop a theory of "deep multisets", *i.e.*, parameterizing universal multiset functions with neural networks. Our next lemma states that sum aggregators can represent injective, in fact, *universal* functions over multisets.

**Lemma 5.** *Assume $\mathcal{X}$ is countable. There exists a function $f : \mathcal{X} \to \mathbb{R}^n$ so that $h(X) = \sum_{x \in X} f(x)$ is unique for each multiset $X \subset \mathcal{X}$ of bounded size. Moreover, any multiset function $g$ can be decomposed as $g(X) = \phi\left(\sum_{x \in X} f(x)\right)$ for some function $\phi$.*

We prove Lemma 5 in the appendix. The proof extends the setting in (Zaheer et al., 2017) from sets to multisets. An important distinction between deep multisets and sets is that certain popular injective set functions, such as the mean aggregator, are not injective multiset functions. With the mechanism for modeling universal multiset functions in Lemma 5 as a building block, we can conceive aggregation schemes that can represent universal functions over a node and the multiset of its neighbors, and thus will satisfy the injectiveness condition (a) in Theorem 3. Our next corollary provides a simple and concrete formulation among many such aggregation schemes.

**Corollary 6.** *Assume $\mathcal{X}$ is countable. There exists a function $f : \mathcal{X} \to \mathbb{R}^n$ so that for infinitely many choices of $\epsilon$, including all irrational numbers, $h(c, X) = (1 + \epsilon) \cdot f(c) + \sum_{x \in X} f(x)$ is unique for each pair $(c, X)$, where $c \in \mathcal{X}$ and $X \subset \mathcal{X}$ is a multiset of bounded size. Moreover, any function $g$ over such pairs can be decomposed as $g(c, X) = \varphi\left((1 + \epsilon) \cdot f(c) + \sum_{x \in X} f(x)\right)$ for some function $\varphi$.*

We can use multi-layer perceptrons (MLPs) to model and learn $f$ and $\varphi$ in Corollary 6, thanks to the universal approximation theorem (Hornik et al., 1989; Hornik, 1991). In practice, we model $f^{(k+1)} \circ \varphi^{(k)}$ with one MLP, because MLPs can represent the composition of functions. In the first iteration, we do not need MLPs before summation if input features are one-hot encodings as their summation alone is injective. We can make $\epsilon$ a learnable parameter or a fixed scalar. Then, GIN updates node representations as

$$h_v^{(k)} = \text{MLP}^{(k)}\left(\left(1 + \epsilon^{(k)}\right) \cdot h_v^{(k-1)} + \sum_{u \in \mathcal{N}(v)} h_u^{(k-1)}\right). \tag{4.1}$$

Generally, there may exist many other powerful GNNs. GIN is one such example among many maximally powerful GNNs, while being simple.

## 4.2 GRAPH-LEVEL READOUT OF GIN

Node embeddings learned by GIN can be directly used for tasks like node classification and link prediction. For graph classification tasks we propose the following "readout" function that, given embeddings of individual nodes, produces the embedding of the entire graph.

An important aspect of the graph-level readout is that node representations, corresponding to subtree structures, get more refined and global as the number of iterations increases. A sufficient number of iterations is key to achieving good discriminative power. Yet, features from earlier iterations may sometimes generalize better. To consider all structural information, we use information from all depths/iterations of the model. We achieve this by an architecture similar to Jumping Knowledge

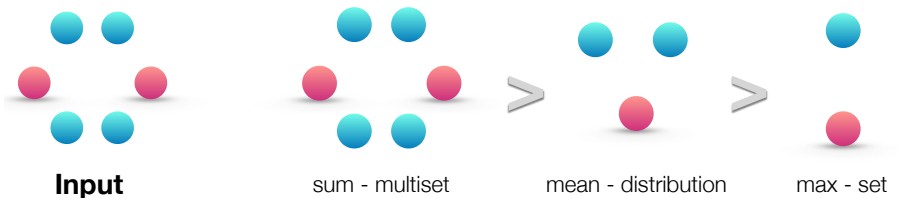

Figure 2: **Ranking by expressive power for sum, mean and max aggregators over a multiset**. Left panel shows the input multiset, *i.e.*, the network neighborhood to be aggregated. The next three panels illustrate the aspects of the multiset a given aggregator is able to capture: sum captures the full multiset, mean captures the proportion/distribution of elements of a given type, and the max aggregator ignores multiplicities (reduces the multiset to a simple set).

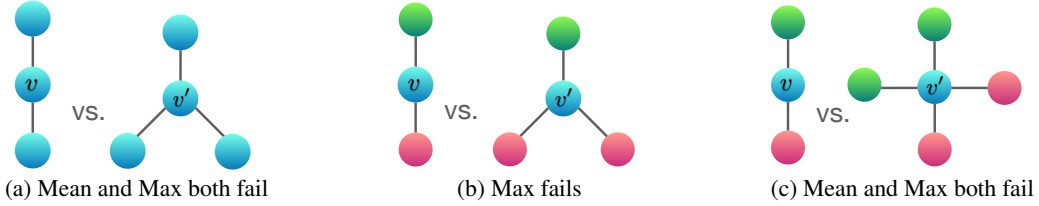

(a) Mean and Max both fail      (b) Max fails      (c) Mean and Max both fail

Figure 3: **Examples of graph structures that mean and max aggregators fail to distinguish.** Between the two graphs, nodes $v$ and $v'$ get the same embedding even though their corresponding graph structures differ. Figure 2 gives reasoning about how different aggregators "compress" different multisets and thus fail to distinguish them.

Networks (Xu et al., 2018), where we replace Eq. 2.4 with graph representations concatenated across *all iterations/layers* of GIN:

$$h_G = \text{CONCAT}\Big(\text{READOUT}\Big(\big\{h_v^{(k)}|v \in G\big\}\Big) \,\big|\, k = 0, 1, \dots, K\Big). \tag{4.2}$$

By Theorem 3 and Corollary 6, if GIN replaces READOUT in Eq. 4.2 with summing all node features from the same iterations (we do not need an extra MLP before summation for the same reason as in Eq. 4.1), it provably generalizes the WL test and the WL subtree kernel.

## 5    LESS POWERFUL BUT STILL INTERESTING GNNS

Next, we study GNNs that do not satisfy the conditions in Theorem 3, including GCN (Kipf & Welling, 2017) and GraphSAGE (Hamilton et al., 2017a). We conduct ablation studies on two aspects of the aggregator in Eq. 4.1: (1) 1-layer perceptrons instead of MLPs and (2) mean or max-pooling instead of the sum. We will see that these GNN variants get confused by surprisingly simple graphs and are less powerful than the WL test. Nonetheless, models with mean aggregators like GCN perform well for *node classification* tasks. To better understand this, we precisely characterize what different GNN variants can and cannot capture about a graph and discuss the implications for learning with graphs.

### 5.1   1-LAYER PERCEPTRONS ARE NOT SUFFICIENT

The function $f$ in Lemma 5 helps map distinct multisets to unique embeddings. It can be parameterized by an MLP by the universal approximation theorem (Hornik, 1991). Nonetheless, many existing GNNs instead use a 1-layer perceptron $\sigma \circ W$ (Duvenaud et al., 2015; Kipf & Welling, 2017; Zhang et al., 2018), a linear mapping followed by a non-linear activation function such as a ReLU. Such 1-layer mappings are examples of Generalized Linear Models (Nelder & Wedderburn, 1972). Therefore, we are interested in understanding whether 1-layer perceptrons are enough for graph learning. Lemma 7 suggests that there are indeed network neighborhoods (multisets) that models with 1-layer perceptrons can never distinguish.

**Lemma 7.** *There exist finite multisets* $X_1 \neq X_2$ *so that for any linear mapping* $W$, $\sum_{x \in X_1} \text{ReLU}(Wx) = \sum_{x \in X_2} \text{ReLU}(Wx).$

The main idea of the proof for Lemma 7 is that 1-layer perceptrons can behave much like linear mappings, so the GNN layers degenerate into simply summing over neighborhood features. Our proof builds on the fact that the bias term is lacking in the linear mapping. With the bias term and sufficiently large output dimensionality, 1-layer perceptrons might be able to distinguish different multisets. Nonetheless, unlike models using MLPs, the 1-layer perceptron (even with the bias term) is *not a universal approximator* of multiset functions. Consequently, even if GNNs with 1-layer perceptrons can embed different graphs to different locations to some degree, such embeddings may not adequately capture structural similarity, and can be difficult for simple classifiers, *e.g.*, linear classifiers, to fit. In Section 7, we will empirically see that GNNs with 1-layer perceptrons, when applied to graph classification, sometimes severely underfit training data and often perform worse than GNNs with MLPs in terms of test accuracy.

## 5.2 STRUCTURES THAT CONFUSE MEAN AND MAX-POOLING

What happens if we replace the sum in $h(X) = \sum_{x \in X} f(x)$ with mean or max-pooling as in GCN and GraphSAGE? Mean and max-pooling aggregators are still well-defined multiset functions because they are permutation invariant. But, they are *not* injective. Figure 2 ranks the three aggregators by their representational power, and Figure 3 illustrates pairs of structures that the mean and max-pooling aggregators fail to distinguish. Here, node colors denote different node features, and we assume the GNNs aggregate neighbors first before combining them with the central node labeled as $v$ and $v'$.

In Figure 3a, every node has the same feature $a$ and $f(a)$ is the same across all nodes (for any function $f$). When performing neighborhood aggregation, the mean or maximum over $f(a)$ remains $f(a)$ and, by induction, we always obtain the same node representation everywhere. Thus, in this case mean and max-pooling aggregators fail to capture any structural information. In contrast, the sum aggregator distinguishes the structures because $2 \cdot f(a)$ and $3 \cdot f(a)$ give different values. The same argument can be applied to any unlabeled graph. If node degrees instead of a constant value is used as node input features, in principle, mean can recover sum, but max-pooling cannot.

Fig. 3a suggests that mean and max have trouble distinguishing graphs with nodes that have repeating features. Let $h_{\text{color}}$ ($r$ for red, $g$ for green) denote node features transformed by $f$. Fig. 3b shows that maximum over the neighborhood of the blue nodes $v$ and $v'$ yields $\max(h_g, h_r)$ and $\max(h_g, h_r, h_r)$, which collapse to the same representation (even though the corresponding graph structures are different). Thus, max-pooling fails to distinguish them. In contrast, the sum aggregator still works because $\frac{1}{2}(h_g + h_r)$ and $\frac{1}{3}(h_g + h_r + h_r)$ are in general not equivalent. Similarly, in Fig. 3c, both mean and max fail as $\frac{1}{2}(h_g + h_r) = \frac{1}{4}(h_g + h_g + h_r + h_r)$.

## 5.3 MEAN LEARNS DISTRIBUTIONS

To characterize the class of multisets that the mean aggregator can distinguish, consider the example $X_1 = (S, m)$ and $X_2 = (S, k \cdot m)$, where $X_1$ and $X_2$ have the same set of distinct elements, but $X_2$ contains $k$ copies of each element of $X_1$. Any mean aggregator maps $X_1$ and $X_2$ to the same embedding, because it simply takes averages over individual element features. Thus, the mean captures the *distribution* (proportions) of elements in a multiset, but not the *exact* multiset.

**Corollary 8.** *Assume $\mathcal{X}$ is countable. There exists a function $f : \mathcal{X} \to \mathbb{R}^n$ so that for $h(X) = \frac{1}{|X|} \sum_{x \in X} f(x)$, $h(X_1) = h(X_2)$ if and only if multisets $X_1$ and $X_2$ have the same distribution. That is, assuming $|X_2| \geq |X_1|$, we have $X_1 = (S, m)$ and $X_2 = (S, k \cdot m)$ for some $k \in \mathbb{N}_{\geq 1}$.*

The mean aggregator may perform well if, for the task, the statistical and distributional information in the graph is more important than the exact structure. Moreover, when node features are diverse and rarely repeat, the mean aggregator is as powerful as the sum aggregator. This may explain why, despite the limitations identified in Section 5.2, GNNs with mean aggregators are effective for node classification tasks, such as classifying article subjects and community detection, where node features are rich and the distribution of the neighborhood features provides a strong signal for the task.

## 5.4 MAX-POOLING LEARNS SETS WITH DISTINCT ELEMENTS

The examples in Figure 3 illustrate that max-pooling considers multiple nodes with the same feature as *only one* node (*i.e.*, treats a multiset as a set). Max-pooling captures neither the exact structure nor

the distribution. However, it may be suitable for tasks where it is important to identify representative elements or the "skeleton", rather than to distinguish the exact structure or distribution. Qi et al. (2017) empirically show that the max-pooling aggregator learns to identify the skeleton of a 3D point cloud and that it is robust to noise and outliers. For completeness, the next corollary shows that the max-pooling aggregator captures the underlying set of a multiset.

**Corollary 9.** *Assume $\mathcal{X}$ is countable. Then there exists a function $f : \mathcal{X} \rightarrow \mathbb{R}^{\infty}$ so that for $h(X) = \max_{x \in X} f(x)$, $h(X_1) = h(X_2)$ if and only if $X_1$ and $X_2$ have the same underlying set.*

### 5.5 REMARKS ON OTHER AGGREGATORS

There are other non-standard neighbor aggregation schemes that we do not cover, *e.g.*, weighted average via attention (Velickovic et al., 2018) and LSTM pooling (Hamilton et al., 2017a; Murphy et al., 2018). We emphasize that our theoretical framework is general enough to characterize the representaional power of any aggregation-based GNNs. In the future, it would be interesting to apply our framework to analyze and understand other aggregation schemes.

## 6 OTHER RELATED WORK

Despite the empirical success of GNNs, there has been relatively little work that mathematically studies their properties. An exception to this is the work of Scarselli et al. (2009a) who shows that the perhaps earliest GNN model (Scarselli et al., 2009b) can approximate measurable functions in probability. Lei et al. (2017) show that their proposed architecture lies in the RKHS of graph kernels, but do not study explicitly which graphs it can distinguish. Each of these works focuses on a specific architecture and do not easily generalize to multple architectures. In contrast, our results above provide a general framework for analyzing and characterizing the expressive power of a broad class of GNNs. Recently, many GNN-based architectures have been proposed, including sum aggregation and MLP encoding (Battaglia et al., 2016; Scarselli et al., 2009b; Duvenaud et al., 2015), and most without theoretical derivation. In contrast to many prior GNN architectures, our Graph Isomorphism Network (GIN) is theoretically motivated, simple yet powerful.

## 7 EXPERIMENTS

We evaluate and compare the training and test performance of GIN and less powerful GNN variants.[1] Training set performance allows us to compare different GNN models based on their representational power and test set performance quantifies generalization ability.

**Datasets.** We use 9 graph classification benchmarks: 4 bioinformatics datasets (MUTAG, PTC, NCI1, PROTEINS) and 5 social network datasets (COLLAB, IMDB-BINARY, IMDB-MULTI, REDDIT-BINARY and REDDIT-MULTI5K) (Yanardag & Vishwanathan, 2015). Importantly, our goal here is not to allow the models to rely on the input node features but mainly learn from the network structure. Thus, in the bioinformatic graphs, the nodes have categorical input features but in the social networks, they have no features. For social networks we create node features as follows: for the REDDIT datasets, we set all node feature vectors to be the same (thus, features here are uninformative); for the other social graphs, we use one-hot encodings of node degrees. Dataset statistics are summarized in Table 1, and more details of the data can be found in Appendix I.

**Models and configurations.** We evaluate GINs (Eqs. 4.1 and 4.2) and the less powerful GNN variants. Under the GIN framework, we consider two variants: (1) a GIN that learns $\epsilon$ in Eq. 4.1 by gradient descent, which we call GIN-$\epsilon$, and (2) a simpler (slightly less powerful)[2] GIN, where $\epsilon$ in Eq. 4.1 is fixed to 0, which we call GIN-0. As we will see, GIN-0 shows strong empirical performance: not only does GIN-0 fit training data equally well as GIN-$\epsilon$, it also demonstrates good generalization, slightly but consistently outperforming GIN-$\epsilon$ in terms of test accuracy. For the less powerful GNN variants, we consider architectures that replace the sum in the GIN-0 aggregation with mean or max-pooling[3], or replace MLPs with 1-layer perceptrons, *i.e.*, a linear mapping followed

---

[1]The code is available at `https://github.com/weihua916/powerful-gnns`.

[2]There exist certain (somewhat contrived) graphs that GIN-$\epsilon$ can distinguish but GIN-0 cannot.

[3]For REDDIT-BINARY, REDDIT–MULTI5K, and COLLAB, we did not run experiments for max-pooling due to GPU memory constraints.

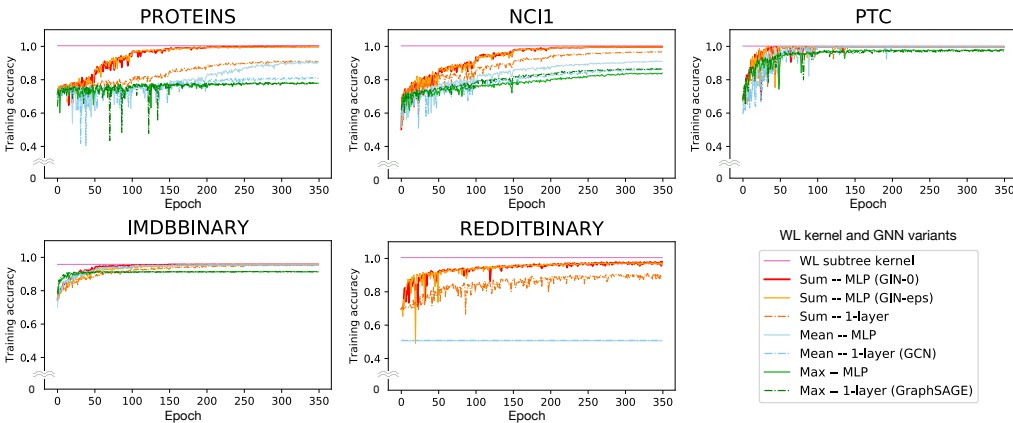

Figure 4: Training set performance of GINs, less powerful GNN variants, and the WL subtree kernel.

by ReLU. In Figure 4 and Table 1, a model is named by the aggregator/perceptron it uses. Here mean–1-layer and max–1-layer correspond to GCN and GraphSAGE, respectively, up to minor architecture modifications. We apply the same graph-level readout (READOUT in Eq. 4.2) for GINs and all the GNN variants, specifically, sum readout on bioinformatics datasets and mean readout on social datasets due to better test performance.

Following (Yanardag & Vishwanathan, 2015; Niepert et al., 2016), we perform 10-fold cross-validation with LIB-SVM (Chang & Lin, 2011). We report the average and standard deviation of validation accuracies across the 10 folds within the cross-validation. For all configurations, 5 GNN layers (including the input layer) are applied, and all MLPs have 2 layers. Batch normalization (Ioffe & Szegedy, 2015) is applied on every hidden layer. We use the Adam optimizer (Kingma & Ba, 2015) with initial learning rate $0.01$ and decay the learning rate by $0.5$ every 50 epochs. The hyper-parameters we tune for each dataset are: (1) the number of hidden units $\in \{16, 32\}$ for bioinformatics graphs and $64$ for social graphs; (2) the batch size $\in \{32, 128\}$; (3) the dropout ratio $\in \{0, 0.5\}$ after the dense layer (Srivastava et al., 2014); (4) the number of epochs, *i.e.*, a single epoch with the best cross-validation accuracy averaged over the 10 folds was selected. Note that due to the small dataset sizes, an alternative setting, where hyper-parameter selection is done using a validation set, is extremely unstable, *e.g.*, for MUTAG, the validation set only contains 18 data points. We also report the training accuracy of different GNNs, where all the hyper-parameters were fixed across the datasets: 5 GNN layers (including the input layer), hidden units of size 64, minibatch of size 128, and 0.5 dropout ratio. For comparison, the training accuracy of the WL subtree kernel is reported, where we set the number of iterations to 4, which is comparable to the 5 GNN layers.

**Baselines.** We compare the GNNs above with a number of state-of-the-art baselines for graph classification: (1) the WL subtree kernel (Shervashidze et al., 2011), where $C$-SVM (Chang & Lin, 2011) was used as a classifier; the hyper-parameters we tune are $C$ of the SVM and the number of WL iterations $\in \{1, 2, \ldots, 6\}$; (2) state-of-the-art deep learning architectures, i.e., Diffusion-convolutional neural networks (DCNN) (Atwood & Towsley, 2016), PATCHY-SAN (Niepert et al., 2016) and Deep Graph CNN (DGCNN) (Zhang et al., 2018); (3) Anonymous Walk Embeddings (AWL) (Ivanov & Burnaev, 2018). For the deep learning methods and AWL, we report the accuracies reported in the original papers.

## 7.1 RESULTS

**Training set performance.** We validate our theoretical analysis of the representational power of GNNs by comparing their training accuracies. Models with higher representational power should have higher training set accuracy. Figure 4 shows training curves of GINs and less powerful GNN variants with the same hyper-parameter settings. First, both the theoretically most powerful GNN, *i.e.* GIN-$\epsilon$ and GIN-0, are able to almost perfectly fit all the training sets. In our experiments, explicit learning of $\epsilon$ in GIN-$\epsilon$ yields no gain in fitting training data compared to fixing $\epsilon$ to 0 as in GIN-0. In comparison, the GNN variants using mean/max pooling or 1-layer perceptrons severely underfit on many datasets. In particular, the training accuracy patterns align with our ranking by the models'

| | Datasets | IMDB-B | IMDB-M | RDT-B | RDT-M5K | COLLAB | MUTAG | PROTEINS | PTC | NCI1 |
|---|---|---|---|---|---|---|---|---|---|---|
| Datasets | # graphs | 1000 | 1500 | 2000 | 5000 | 5000 | 188 | 1113 | 344 | 4110 |
| | # classes | 2 | 3 | 2 | 5 | 3 | 2 | 2 | 2 | 2 |
| | Avg # nodes | 19.8 | 13.0 | 429.6 | 508.5 | 74.5 | 17.9 | 39.1 | 25.5 | 29.8 |
| Baselines | WL subtree | 73.8 ± 3.9 | 50.9 ± 3.8 | 81.0 ± 3.1 | 52.5 ± 2.1 | 78.9 ± 1.9 | 90.4 ± 5.7 | 75.0 ± 3.1 | 59.9 ± 4.3 | **86.0 ± 1.8** * |
| | DCNN | 49.1 | 33.5 | – | – | 52.1 | 67.0 | 61.3 | 56.6 | 62.6 |
| | PATCHYSAN | 71.0 ± 2.2 | 45.2 ± 2.8 | 86.3 ± 1.6 | 49.1 ± 0.7 | 72.6 ± 2.2 | **92.6 ± 4.2** * | 75.9 ± 2.8 | 60.0 ± 4.8 | 78.6 ± 1.9 |
| | DGCNN | 70.0 | 47.8 | – | – | 73.7 | 85.8 | 75.5 | 58.6 | 74.4 |
| | AWL | 74.5 ± 5.9 | 51.5 ± 3.6 | 87.9 ± 2.5 | 54.7 ± 2.9 | 73.9 ± 1.9 | 87.9 ± 9.8 | – | – | – |
| GNN variants | SUM–MLP (**GIN-0**) | **75.1 ± 5.1** | **52.3 ± 2.8** | **92.4 ± 2.5** | **57.5 ± 1.5** | **80.2 ± 1.9** | **89.4 ± 5.6** | **76.2 ± 2.8** | **64.6 ± 7.0** | **82.7 ± 1.7** |
| | SUM–MLP (**GIN-$\epsilon$**) | **74.3 ± 5.1** | **52.1 ± 3.6** | **92.2 ± 2.3** | **57.0 ± 1.7** | **80.1 ± 1.9** | **89.0 ± 6.0** | **75.9 ± 3.8** | 63.7 ± 8.2 | **82.7 ± 1.6** |
| | SUM–1-LAYER | 74.1 ± 5.0 | **52.2 ± 2.4** | **90.0 ± 2.7** | 55.1 ± 1.6 | **80.6 ± 1.9** | **90.0 ± 8.8** | **76.2 ± 2.6** | 63.1 ± 5.7 | 82.0 ± 1.5 |
| | MEAN–MLP | 73.7 ± 3.7 | **52.3 ± 3.1** | 50.0 ± 0.0 | 20.0 ± 0.0 | 79.2 ± 2.3 | 83.5 ± 6.3 | 75.5 ± 3.4 | **66.6 ± 6.9** | 80.9 ± 1.8 |
| | MEAN–1-LAYER (GCN) | 74.0 ± 3.4 | 51.9 ± 3.8 | 50.0 ± 0.0 | 20.0 ± 0.0 | 79.0 ± 1.8 | 85.6 ± 5.8 | 76.0 ± 3.2 | 64.2 ± 4.3 | 80.2 ± 2.0 |
| | MAX–MLP | 73.2 ± 5.8 | 51.1 ± 3.6 | – | – | – | 84.0 ± 6.1 | 76.0 ± 3.2 | 64.6 ± 10.2 | 77.8 ± 1.3 |
| | MAX–1-LAYER (GraphSAGE) | 72.3 ± 5.3 | 50.9 ± 2.2 | – | – | – | 85.1 ± 7.6 | 75.9 ± 3.2 | 63.9 ± 7.7 | 77.7 ± 1.5 |

Table 1: **Test set classification accuracies (%).** The best-performing GNNs are highlighted with boldface. On datasets where GINs' accuracy is not strictly the highest among GNN variants, we see that GINs are still comparable to the best GNN because a paired t-test at significance level 10% does not distinguish GINs from the best; thus, GINs are also highlighted with boldface. If a baseline performs significantly better than all GNNs, we highlight it with boldface and asterisk.

representational power: GNN variants with MLPs tend to have higher training accuracies than those with 1-layer perceptrons, and GNNs with sum aggregators tend to fit the training sets better than those with mean and max-pooling aggregators.

On our datasets, training accuracies of the GNNs never exceed those of the WL subtree kernel. This is expected because GNNs generally have lower discriminative power than the WL test. For example, on IMDBBINARY, none of the models can perfectly fit the training set, and the GNNs achieve at most the same training accuracy as the WL kernel. This pattern aligns with our result that the WL test provides an upper bound for the representational capacity of the aggregation-based GNNs. However, the WL kernel is not able to learn how to combine node features, which might be quite informative for a given prediction task as we will see next.

**Test set performance.** Next, we compare test accuracies. Although our theoretical results do not directly speak about the generalization ability of GNNs, it is reasonable to expect that GNNs with strong expressive power can accurately capture graph structures of interest and thus generalize well. Table 1 compares test accuracies of GINs (Sum–MLP), other GNN variants, as well as the state-of-the-art baselines.

First, GINs, especially GIN-0, outperform (or achieve comparable performance as) the less powerful GNN variants on all the 9 datasets, achieving state-of-the-art performance. GINs shine on the social network datasets, which contain a relatively large number of training graphs. For the Reddit datasets, all nodes share the same scalar as node feature. Here, GINs and sum-aggregation GNNs accurately capture the graph structure and significantly outperform other models. Mean-aggregation GNNs, however, fail to capture any structures of the unlabeled graphs (as predicted in Section 5.2) and do not perform better than random guessing. Even if node degrees are provided as input features, mean-based GNNs perform much worse than sum-based GNNs (the accuracy of the GNN with mean–MLP aggregation is 71.2±4.6% on REDDIT-BINARY and 41.3±2.1% on REDDIT-MULTI5K). Comparing GINs (GIN-0 and GIN-$\epsilon$), we observe that GIN-0 slightly but consistently outperforms GIN-$\epsilon$. Since both models fit training data equally well, the better generalization of GIN-0 may be explained by its simplicity compared to GIN-$\epsilon$.

## 8 CONCLUSION

In this paper, we developed theoretical foundations for reasoning about the expressive power of GNNs, and proved tight bounds on the representational capacity of popular GNN variants. We also designed a provably maximally powerful GNN under the neighborhood aggregation framework. An interesting direction for future work is to go beyond neighborhood aggregation (or message passing) in order to pursue possibly even more powerful architectures for learning with graphs. To complete the picture, it would also be interesting to understand and improve the generalization properties of GNNs as well as better understand their optimization landscape.

ACKNOWLEDGMENTS

This research was supported by NSF CAREER award 1553284, a DARPA D3M award and DARPA DSO's Lagrange program under grant FA86501827838. This research was also supported in part by NSF, ARO MURI, Boeing, Huawei, Stanford Data Science Initiative, and Chan Zuckerberg Biohub. Weihua Hu was supported by Funai Overseas Scholarship. We thank Prof. Ken-ichi Kawarabayashi and Prof. Masashi Sugiyama for supporting this research with computing resources and providing great advice. We thank Tomohiro Sonobe and Kento Nozawa for managing servers. We thank Rex Ying and William Hamilton for helpful feedback. We thank Simon S. Du, Yasuo Tabei, Chengtao Li, and Jingling Li for helpful discussions and positive comments.

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

## A  PROOF FOR LEMMA 2

*Proof.* Suppose after $k$ iterations, a graph neural network $\mathcal{A}$ has $\mathcal{A}(G_1) \neq \mathcal{A}(G_2)$ but the WL test cannot decide $G_1$ and $G_2$ are non-isomorphic. It follows that from iteration 0 to $k$ in the WL test, $G_1$ and $G_2$ always have the same collection of node labels. In particular, because $G_1$ and $G_2$ have the same WL node labels for iteration $i$ and $i + 1$ for any $i = 0, ..., k - 1$, $G_1$ and $G_2$ have the same collection, i.e. multiset, of WL node labels $\left\{ l_v^{(i)} \right\}$ as well as the same collection of node neighborhoods $\left\{ \left( l_v^{(i)}, \left\{ l_u^{(i)} : u \in \mathcal{N}(v) \right\} \right) \right\}$. Otherwise, the WL test would have obtained different collections of node labels at iteration $i + 1$ for $G_1$ and $G_2$ as different multisets get unique new labels.

The WL test always relabels different multisets of neighboring nodes into different new labels. We show that on the same graph $G = G_1$ or $G_2$, if WL node labels $l_v^{(i)} = l_u^{(i)}$, we always have GNN node features $h_v^{(i)} = h_u^{(i)}$ for any iteration $i$. This apparently holds for $i = 0$ because WL and GNN starts with the same node features. Suppose this holds for iteration $j$, if for any $u, v$, $l_v^{(j+1)} = l_u^{(j+1)}$, then it must be the case that

$$\left( l_v^{(j)}, \left\{ l_w^{(j)} : w \in \mathcal{N}(v) \right\} \right) = \left( l_u^{(j)}, \left\{ l_w^{(j)} : w \in \mathcal{N}(u) \right\} \right)$$

By our assumption on iteration $j$, we must have

$$\left( h_v^{(j)}, \left\{ h_w^{(j)} : w \in \mathcal{N}(v) \right\} \right) = \left( h_u^{(j)}, \left\{ h_w^{(j)} : w \in \mathcal{N}(u) \right\} \right)$$

In the aggregation process of the GNN, the same AGGREGATE and COMBINE are applied. The same input, i.e. neighborhood features, generates the same output. Thus, $h_v^{(j+1)} = h_u^{(j+1)}$. By induction, if WL node labels $l_v^{(i)} = l_u^{(i)}$, we always have GNN node features $h_v^{(i)} = h_u^{(i)}$ for any iteration $i$. This creates a valid mapping $\phi$ such that $h_v^{(i)} = \phi(l_v^{(i)})$ for any $v \in G$. It follows from $G_1$ and $G_2$ have the same multiset of WL neighborhood labels that $G_1$ and $G_2$ also have the same collection of GNN neighborhood features

$$\left\{ \left( h_v^{(i)}, \left\{ h_u^{(i)} : u \in \mathcal{N}(v) \right\} \right) \right\} = \left\{ \left( \phi(l_v^{(i)}), \left\{ \phi(l_u^{(i)}) : u \in \mathcal{N}(v) \right\} \right) \right\}$$

Thus, $\left\{ h_v^{(i+1)} \right\}$ are the same. In particular, we have the same collection of GNN node features $\left\{ h_v^{(k)} \right\}$ for $G_1$ and $G_2$. Because the graph level readout function is permutation invariant with respect to the collection of node features, $\mathcal{A}(G_1) = \mathcal{A}(G_2)$. Hence we have reached a contradiction.  □

## B  PROOF FOR THEOREM 3

*Proof.* Let $\mathcal{A}$ be a graph neural network where the condition holds. Let $G_1, G_2$ be any graphs which the WL test decides as non-isomorphic at iteration $K$. Because the graph-level readout function is injective, i.e., it maps distinct multiset of node features into unique embeddings, it suffices to show that $\mathcal{A}$'s neighborhood aggregation process, with sufficient iterations, embeds $G_1$ and $G_2$ into different multisets of node features. Let us assume $\mathcal{A}$ updates node representations as

$$h_v^{(k)} = \phi \left( h_v^{(k-1)}, f \left( \left\{ h_u^{(k-1)} : u \in \mathcal{N}(v) \right\} \right) \right)$$

with injective funtions $f$ and $\phi$. The WL test applies a predetermined injective hash function $g$ to update the WL node labels $l_v^{(k)}$:

$$l_v^{(k)} = g \left( l_v^{(k-1)}, \left\{ l_u^{(k-1)} : u \in \mathcal{N}(v) \right\} \right)$$

We will show, by induction, that for any iteration $k$, there always exists an injective function $\varphi$ such that $h_v^{(k)} = \varphi \left( l_v^{(k)} \right)$. This apparently holds for $k = 0$ because the initial node features are the same

for WL and GNN $l_v^{(0)} = h_v^{(0)}$ for all $v \in G_1, G_2$. So $\varphi$ could be the identity function for $k = 0$. Suppose this holds for iteration $k - 1$, we show that it also holds for $k$. Substituting $h_v^{(k-1)}$ with $\varphi\left(l_v^{(k-1)}\right)$ gives us

$$h_v^{(k)} = \phi\left(\varphi\left(l_v^{(k-1)}\right), f\left(\left\{\varphi\left(l_u^{(k-1)}\right) : u \in \mathcal{N}(v)\right\}\right)\right).$$

Since the composition of injective functions is injective, there exists some injective function $\psi$ so that

$$h_v^{(k)} = \psi\left(l_v^{(k-1)}, \left\{l_u^{(k-1)} : u \in \mathcal{N}(v)\right\}\right)$$

Then we have

$$h_v^{(k)} = \psi \circ g^{-1} g\left(l_v^{(k-1)}, \left\{l_u^{(k-1)} : u \in \mathcal{N}(v)\right\}\right) = \psi \circ g^{-1}\left(l_v^{(k)}\right)$$

$\varphi = \psi \circ g^{-1}$ is injective because the composition of injective functions is injective. Hence for any iteration $k$, there always exists an injective function $\varphi$ such that $h_v^{(k)} = \varphi\left(l_v^{(k)}\right)$. At the $K$-th iteration, the WL test decides that $G_1$ and $G_2$ are non-isomorphic, that is the multisets $\left\{l_v^{(K)}\right\}$ are different for $G_1$ and $G_2$. The graph neural network $\mathcal{A}$'s node embeddings $\left\{h_v^{(K)}\right\} = \left\{\varphi\left(l_v^{(K)}\right)\right\}$ must also be different for $G_1$ and $G_2$ because of the injectivity of $\varphi$.

$\square$

## C    Proof for Lemma 4

*Proof.* Before proving our lemma, we first show a well-known result that we will later reduce our problem to: $\mathbb{N}^k$ is countable for every $k \in \mathbb{N}$, i.e. finite Cartesian product of countable sets is countable. We observe that it suffices to show $\mathbb{N} \times \mathbb{N}$ is countable, because the proof then follows clearly from induction. To show $\mathbb{N} \times \mathbb{N}$ is countable, we construct a bijection $\phi$ from $\mathbb{N} \times \mathbb{N}$ to $\mathbb{N}$ as

$$\phi\left(m, n\right) = 2^{m-1} \cdot (2n - 1)$$

Now we go back to proving our lemma. If we can show that the range of any function $g$ defined on multisets of bounded size from a countable set is also countable, then the lemma holds for any $g^{(k)}$ by induction. Thus, our goal is to show that the range of such $g$ is countable. First, it is clear that the mapping from $g(X)$ to $X$ is injective because $g$ is a well-defined function. It follows that it suffices to show the set of all multisets $X \subset \mathcal{X}$ is countable.

Since the union of two countable sets is countable, the following set $\mathcal{X}'$ is also countable.

$$\mathcal{X}' = \mathcal{X} \cup \{e\}$$

where $e$ is a dummy element that is not in $\mathcal{X}$. It follows from the result we showed above, *i.e.*, $\mathbb{N}^k$ is countable for every $k \in \mathbb{N}$, that $\mathcal{X}'^k$ is countable for every $k \in \mathbb{N}$. It remains to show there exists an injective mapping from the set of multisets in $\mathcal{X}$ to $\mathcal{X}'^k$ for some $k \in \mathbb{N}$.

We construct an injective mapping $h$ from the set of multisets $X \subset \mathcal{X}$ to $\mathcal{X}'^k$ for some $k \in \mathbb{N}$ as follows. Because $\mathcal{X}$ is countable, there exists a mapping $Z : \mathcal{X} \to \mathbb{N}$ from $x \in \mathcal{X}$ to natural numbers. We can sort the elements $x \in X$ by $z(x)$ as $x_1, x_2, ..., x_n$, where $n = |X|$. Because the multisets $X$ are of bounded size, there exists $k \in \mathbb{N}$ so that $|X| < k$ for all $X$. We can then define $h$ as

$$h\left(X\right) = (x_1, x_2, ..., x_n, e, e, e...),$$

where the $k - n$ coordinates are filled with the dummy element $e$. It is clear that $h$ is injective because for any multisets $X$ and $Y$ of bounded size, $h(X) = h(Y)$ only if $X$ is equivalent to $Y$. Hence it follows that the range of $g$ is countable as desired.

$\square$

## D  PROOF FOR LEMMA 5

*Proof.* We first prove that there exists a mapping $f$ so that $\sum_{x \in X} f(x)$ is unique for each multiset $X$ of bounded size. Because $\mathcal{X}$ is countable, there exists a mapping $Z : \mathcal{X} \to \mathbb{N}$ from $x \in \mathcal{X}$ to natural numbers. Because the cardinality of multisets $X$ is bounded, there exists a number $N \in \mathbb{N}$ so that $|X| < N$ for all $X$. Then an example of such $f$ is $f(x) = N^{-Z(x)}$. This $f$ can be viewed as a more compressed form of an one-hot vector or $N$-digit presentation. Thus, $h(X) = \sum_{x \in X} f(x)$ is an injective function of multisets.

$\phi \left( \sum_{x \in X} f(x) \right)$ is permutation invariant so it is a well-defined multiset function. For any multiset function $g$, we can construct such $\phi$ by letting $\phi \left( \sum_{x \in X} f(x) \right) = g(X)$. Note that such $\phi$ is well-defined because $h(X) = \sum_{x \in X} f(x)$ is injective.

$\square$

## E  PROOF OF COROLLARY 6

*Proof.* Following the proof of Lemma 5, we consider $f(x) = N^{-Z(x)}$, where $N$ and $Z : \mathcal{X} \to \mathbb{N}$ are the same as defined in Appendix D. Let $h(c, X) \equiv (1 + \epsilon) \cdot f(c) + \sum_{x \in X} f(x)$. Our goal is show that for any $(c', X') \neq (c, X)$ with $c, c' \in \mathcal{X}$ and $X, X' \subset \mathcal{X}$, $h(c, X) \neq h(c', X')$ holds, if $\epsilon$ is an irrational number. We prove by contradiction. For any $(c, X)$, suppose there exists $(c', X')$ such that $(c', X') \neq (c, X)$ but $h(c, X) = h(c', X')$ holds. Let us consider the following two cases: (1) $c' = c$ but $X' \neq X$, and (2) $c' \neq c$. For the first case, $h(c, X) = h(c, X')$ implies $\sum_{x \in X} f(x) = \sum_{x \in X'} f(x)$. It follows from Lemma 5 that the equality will not hold, because with $f(x) = N^{-Z(x)}$, $X' \neq X$ implies $\sum_{x \in X} f(x) \neq \sum_{x \in X'} f(x)$. Thus, we reach a contradiction. For the second case, we can similarly rewrite $h(c, X) = h(c', X')$ as

$$\epsilon \cdot (f(c) - f(c')) = \left( f(c') + \sum_{x \in X'} f(x) \right) - \left( f(c) + \sum_{x \in X} f(x) \right). \tag{E.1}$$

Because $\epsilon$ is an irrational number and $f(c) - f(c')$ is a non-zero rational number, L.H.S. of Eq. E.1 is irrational. On the other hand, R.H.S. of Eq. E.1, the sum of a finite number of rational numbers, is rational. Hence the equality in Eq. E.1 cannot hold, and we have reached a contradiction.

For any function $g$ over the pairs $(c, X)$, we can construct such $\varphi$ for the desired decomposition by letting $\varphi \left( (1 + \epsilon) \cdot f(c) + \sum_{x \in X} f(x) \right) = g(c, X)$. Note that such $\varphi$ is well-defined because $h(c, X) = (1 + \epsilon) \cdot f(c) + \sum_{x \in X} f(x)$ is injective. $\square$

## F  PROOF FOR LEMMA 7

*Proof.* Let us consider the example $X_1 = \{1, 1, 1, 1, 1\}$ and $X_2 = \{2, 3\}$, *i.e.* two different multisets of positive numbers that sum up to the same value. We will be using the homogeneity of ReLU.

Let $W$ be an arbitrary linear transform that maps $x \in X_1, X_2$ into $\mathbb{R}^n$. It is clear that, at the same coordinates, $Wx$ are either positive or negative for all $x$ because all $x$ in $X_1$ and $X_2$ are positive. It follows that ReLU$(Wx)$ are either positive or 0 at the same coordinate for all $x$ in $X_1, X_2$. For the coordinates where ReLU$(Wx)$ are 0, we have $\sum_{x \in X_1}$ ReLU $(Wx) = \sum_{x \in X_2}$ ReLU $(Wx)$. For the coordinates where $Wx$ are positive, linearity still holds. It follows from linearity that

$$\sum_{x \in X} \text{ReLU} (Wx) = \text{ReLU} \left( W \sum_{x \in X} x \right)$$

where $X$ could be $X_1$ or $X_2$. Because $\sum_{x \in X_1} x = \sum_{x \in X_2} x$, we have the following as desired.

$$\sum_{x \in X_1} \text{ReLU} (Wx) = \sum_{x \in X_2} \text{ReLU} (Wx)$$

$\square$

## G    PROOF FOR COROLLARY 8

*Proof.* Suppose multisets $X_1$ and $X_2$ have the same distribution, without loss of generality, let us assume $X_1 = (S, m)$ and $X_2 = (S, k \cdot m)$ for some $k \in \mathbb{N}_{\geq 1}$, i.e. $X_1$ and $X_2$ have the same underlying set and the multiplicity of each element in $X_2$ is $k$ times of that in $X_1$. Then we have $|X_2| = k|X_1|$ and $\sum_{x \in X_2} f(x) = k \cdot \sum_{x \in X_1} f(x)$. Thus,

$$\frac{1}{|X_2|} \sum_{x \in X_2} f(x) = \frac{1}{k \cdot |X_1|} \cdot k \cdot \sum_{x \in X_1} f(x) = \frac{1}{|X_1|} \sum_{x \in X_1} f(x)$$

Now we show that there exists a function $f$ so that $\frac{1}{|X|} \sum_{x \in X} f(x)$ is unique for distributionally equivalent $X$. Because $\mathcal{X}$ is countable, there exists a mapping $Z : \mathcal{X} \to \mathbb{N}$ from $x \in \mathcal{X}$ to natural numbers. Because the cardinality of multisets $X$ is bounded, there exists a number $N \in \mathbb{N}$ so that $|X| < N$ for all $X$. Then an example of such $f$ is $f(x) = N^{-2Z(x)}$. □

## H    PROOF FOR COROLLARY 9

*Proof.* Suppose multisets $X_1$ and $X_2$ have the same underlying set $S$, then we have

$$\max_{x \in X_1} f(x) = \max_{x \in S} f(x) = \max_{x \in X_2} f(x)$$

Now we show that there exists a mapping $f$ so that $\max_{x \in X} f(x)$ is unique for $X$s with the same underlying set. Because $\mathcal{X}$ is countable, there exists a mapping $Z : \mathcal{X} \to \mathbb{N}$ from $x \in \mathcal{X}$ to natural numbers. Then an example of such $f : \mathcal{X} \to \mathbb{R}^\infty$ is defined as $f_i(x) = 1$ for $i = Z(x)$ and $f_i(x) = 0$ otherwise, where $f_i(x)$ is the $i$-th coordinate of $f(x)$. Such an $f$ essentially maps a multiset to its one-hot embedding. □

## I    DETAILS OF DATASETS

We give detailed descriptions of datasets used in our experiments. Further details can be found in (Yanardag & Vishwanathan, 2015).

**Social networks datasets.**    IMDB-BINARY and IMDB-MULTI are movie collaboration datasets. Each graph corresponds to an ego-network for each actor/actress, where nodes correspond to actors/actresses and an edge is drawn betwen two actors/actresses if they appear in the same movie. Each graph is derived from a pre-specified genre of movies, and the task is to classify the genre graph it is derived from. REDDIT-BINARY and REDDIT-MULTI5K are balanced datasets where each graph corresponds to an online discussion thread and nodes correspond to users. An edge was drawn between two nodes if at least one of them responded to another's comment. The task is to classify each graph to a community or a subreddit it belongs to. COLLAB is a scientific collaboration dataset, derived from 3 public collaboration datasets, namely, High Energy Physics, Condensed Matter Physics and Astro Physics. Each graph corresponds to an ego-network of different researchers from each field. The task is to classify each graph to a field the corresponding researcher belongs to.

**Bioinformatics datasets.**    MUTAG is a dataset of 188 mutagenic aromatic and heteroaromatic nitro compounds with 7 discrete labels. PROTEINS is a dataset where nodes are secondary structure elements (SSEs) and there is an edge between two nodes if they are neighbors in the amino-acid sequence or in 3D space. It has 3 discrete labels, representing helix, sheet or turn. PTC is a dataset of 344 chemical compounds that reports the carcinogenicity for male and female rats and it has 19 discrete labels. NCI1 is a dataset made publicly available by the National Cancer Institute (NCI) and is a subset of balanced datasets of chemical compounds screened for ability to suppress or inhibit the growth of a panel of human tumor cell lines, having 37 discrete labels.

