# OpenReview forum: "How Powerful are Graph Neural Networks?"
_ICLR.cc/2019/Conference_

### Official Review · AnonReviewer3 · 2018-11-01
**Reviewer comment**

**Rating:** 8
**Confidence:** 5

**Review:**

This paper presents a very interesting investigation of the expressive capabilities of graph neural networks, in particular focusing on the discriminative power of such GNN models, i.e. the ability to tell that two inputs are different when they are actually different.  The analysis is based on the study of injective representation functions on multisets.  This perspective in particular allows the authors to distinguish different aggregation methods, sum, mean and max, as well as to distinguish one layer linear transformations from multi-layer MLPs.  Based on the analysis the authors proposed a variant of the GNN called Graph Isomorphism Networks (GINs) that use MLPs instead of linear transformations on each layer, and sum instead of mean or max as the aggregation method, which has the most discriminative power following the analysis.  Experiments were done on node classification benchmarks to support the claims.

Overall I quite liked this paper.  The study of the expressive capabilities of GNNs is a very important problem.  Given the popularity of this class of models recently, theoretical analysis for these models is largely missing.  Previous attempts at studying the capability of GNNs focus on the function approximation perspective (e.g. Mapping Images to Scene Graphs with Permutation-Invariant Structured Prediction by Hertiz et al. which is worth discussing).  This paper presents a very different angle focusing on discriminative capabilities.  Being able to tell two inputs apart when they are different is obviously just one aspect of representation power, but this paper showed that studying this aspect can already give us some interesting insights.

I do feel however that the authors should make it clear that discriminative power is not the only thing we care, and in most applications we are not doing graph isomorphism tests.  The ability to tell, for example, how far two inputs are, when they are not the same is also very (and maybe more) important, which such isomorphism / injective map based analysis does not capture at all.  In fact the assumption that each feature vector can be mapped to a unique label in {a, b, c, ...} (Section 3 first paragraph) is overly simplistic and only makes sense for analyzing injective maps.  If we want to reason anything about the continuity of the features and representations, this assumption does not apply, and the real set is not countable so such a mapping cannot exist.

In equation 4.1 describes the GIN update, which is proposed as “the most powerful GNN”.  However, such architecture is not really new, for example the Interaction Networks (Battaglia et al. 2016) already uses sum aggregation and MLP as the building blocks.  Also, it is said that in the first iteration a simple sum is enough to implement injective map, this is true for sum, but replacing that with mean and max can lose information very early on.  Another MLP on the input features at least for mean or max aggregation for the first iteration is therefore necessary.  This isn’t made very clear in the paper.

The training set results presented in section 6.1 is not very clear.  The plots show only one run for each model variant, which run was it?  As the purpose is to show that some variants fit well, and some others overfit, these runs should be chosen to optimize training set performance, rather than generalization.  Also the restrictions should be made clear that all models are given the same (small) amount of hidden units per node.  I imagine if the amount of hidden units are allowed to be much bigger, mean and max aggregators should also catch up.

As mentioned earlier I quite liked the paper despite some restrictions anc things to clarify.  I would vote for accepting this paper for publication at ICLR.

--------

Considering the counter-example given above, I'm lowering my scores a bit.  The proof of theorem 3 is less than clear.  The proof for the first half of theorem 3 (a) is quite obvious, but the proof for the second half is a bit hand-wavy.

In the worst case, the second half of theorem 3 (a) will be invalid.  The most general GNN will then have to use an update function in the form of the first half of 3(a), and all the other analysis still holds.  The experiments will need to be rerun.

--------

Update: the new revision resolved the counter-example issue and I'm mostly happy with it, so my rating was adjusted again.

---

> ### Author Response · Authors · 2018-11-22
> **Response to Reviewer3**
>
> We thank the reviewer for the positive review and constructive feedback! We are glad that the reviewer likes our paper.
>
> First, we completely agree that the ability of GNNs to capture structural similarity of graphs is very important besides their discriminative power, and we believe this is one of the most important benefits of using GNNs over WL kernel. We have now made this point clearer in Section 4. Furthermore, we emphasized that we do consider node features to lie in R^d so that they can capture the similarity. The subtlety is that (as R1 nicely pointed out), we need a common assumption that node features at each layer are from countable set in R^d (not from the entire R^d). This is satisfied if the input node features are from a countable set, because for a graph neural network, countability propagates across all layers in a GNN. We leave uncountable node input features for future work and add a more detailed discussion in Section 4 of the revised paper.
>
> In the following, we respond to R3’s other helpful comments and suggestions:
>
> 1. RE: Architecture is similar to, e.g., Interaction Networks
> Thank you for the pointers. Some of our GIN’s building blocks, e.g. sum and MLP indeed appeared in other architectures. We emphasize that while previous work tend to be somewhat ad-hoc in designing GNN architectures, our main emphasis is on deriving our GIN architecture based on the theoretical motivation. In Section 6 of the revised version, we mention related GNN architectures and discuss the differences.
>
> 2. RE: Using MLP for mean or max in the initial step is more fair?
> We think there might be a slight misunderstanding here: as we discussed with concrete examples in Section 5.2, mean or max pooling are inherently incapable of capturing the multiset information regardless of the use of MLP. Especially, in our experiments, we use one-hot encodings as input node features, so the use of MLP on top of them does not increase the discriminative power of mean/max pooling.
>
> 3. RE: Training set results optimized for test performance?
> The results were not actually optimized for test performance. Instead, we used exactly the same configurations for all the datasets: For all the GNNs, the same configurations were used across datasets: 5 GNN layers (including the input layer), hidden units of size 64, minibatch of size 128, and 0.5 dropout ratio. For the WL subtree kernel, we set the number of iterations to 4, which is comparable to the 5 GNN layers. We clarified this in Figure 6 of the revised paper.

---

> > ### Comment · AnonReviewer3 · 2018-11-29
> > **Thanks for the response.**
> >
> > I thank the authors for the revision of the paper and the response.   I have readjusted my rating.
> >
> > The solution to the question raised by the counter example in the new equation (4.1) is a technical one, I would rather prefer not to simplify the function g(c, X) which uses two functions phi and f in this form, as it really doesn't buy us much.
> >
> > W.r.t. related work, the statement "Not surprisingly, some building blocks of GIN, e.g. sum aggregation
> > and MLP encoding, also appeared in other models" (section 6) is not fair and misleading.  As it is not the case that "some building blocks" also appear in other models, but rather some other models, like interaction networks, already contains "all" the essential building blocks (sum, MLP, etc.) presented in this paper.  This doesn't undermine the theoretical contribution of this paper, but the authors should be fair to previous work.

---

> > > ### Author Response · Authors · 2018-11-30
> > > **Response to R3**
> > >
> > > Thank you for the encouraging review! We respond to your further comments below.
> > >
> > > 1) We probably do not fully understand your comment regarding Eqn (4.1) and g(c,X). Especially, could you please clarify your meaning of “simplify g(c, X)”? In our GIN in Eqn (4.1), we compose phi and f in Corollary 6.
> > >
> > > 2) We will further edit related work according to your suggestions. Interaction Networks is a great work and we like it.

---

> > > > ### Comment · AnonReviewer3 · 2018-11-30
> > > > **Eq 4.1 and g(c, X)**
> > > >
> > > > What I meant was, in g(c, X) you have two functions phi and f, which is the form required by Theorem 3.  The problem of the counter-example comes in when you used a single function instead of 2 functions, which ignores the difference between the node at the center and all its neighbors.
> > > >  Introducing an epsilon is a technical solution to this problem (in my opinion), I think you actually don't need this because the original form of g(c, X) is enough, and using a single function rather than 2 does not save you much.
> > > >
> > > > Note: I think of phi and f as MLPs, "," as concat, and "{}" as some aggregation operator, like sum.

---

> > > > > ### Author Response · Authors · 2018-11-30
> > > > > **Effectiveness of Eqn (4.1) and GINs**
> > > > >
> > > > > Thank you for the clarification. We would like to first clarify that the letters (phi, f) in Corollary 6 and Theorem 3 do not have direct correspondence; but we can easily rearrange Eqn (4.1) to obtain the corresponding (phi, f) in the form of Theorem 3. Intuitively, what Theorem 3 asks for is to injectively represent a pair of a node and its neighbors, so the injective function, g(c, X), corresponds to (phi, f) in Theorem 3.
> > > > >
> > > > > Furthermore, our motivation for designing Eqn (4.1), i.e. GIN-0 and GIN-eps, rather than simply applying concatenation, is for better empirical performance. In our preliminary experiments, we found such concatenation was harder to train compared to our simple GINs (both GIN-0 and GIN-eps) and achieved lower test accuracy than GINs. The simplicity of GINs brings better performance in practice. We leave the extensive investigation and comparison to our future work.

---

### Official Review · AnonReviewer2 · 2018-11-02
**Nice results on the expressive power of neighborhood aggregation mechanisms used in GNNs**

**Rating:** 7
**Confidence:** 5

**Review:**

The author study the expressive power of neighborhood aggregation mechanisms used in Graph Neural Networks and relates them to the 1-dimensional Weisfeiler-Lehman heuristic (1-WL) for graph isomorphism testing. The authors show that GCNs with injections acting on the neighborhood features can distinguish the same graphs that can be distinguished by 1-WL. Moreover, they propose a simple GNN layer, namely GIN, that satisfies this property. Moreover, less powerful GNN layers are studied, such as GCN or GraphSage. Their advantages and disadvantages are discussed and it is shown which graph structures they can distinguish. Finally, the paper shows that the GIN layer beats SOTA GNN layers on well-known benchmark datasets from the graph kernel literature.

Studying the expressive power of neighborhood aggregation mechanisms is an important contribution to the further development of GCNs. The paper is well-written and easy to follow. The experimental results are well explained and the evaluation is convincing.

However, I have some concerns regarding the main result in Theorem 3. A consequence of the theorem is that it makes no differences (w.r.t. expressive power) whether one distinguishes the features of the node itself from those of its neighbors. This is remarkable and counterintuitive, but not discussed in the article. However, it is discussed in the proof of Theorem 3 (Appendix) which suggests that the number of iterations must be increased for some graphs in order to obtain the same expressive power. Unfortunately, at this point, the proof is a bit vague. I would like to see a discussion of this differences in the article. This should be clarified in a revised version.
----
Edit:
The counter example posted in a comment ( https://openreview.net/forum?id=ryGs6iA5Km&noteId=rkl2Q1Qi6X&noteId=rkl2Q1Qi6X ) actually shows that my concerns regarding Theorem 3 and its proof were perfectly justified. I agree that the two graphs provide a counterexample to the main result of the paper. Therefore, I have adjusted my rating. I will increase my rating again when the problem can be resolved. However, this appears to be non-trivial.
----
Moreover, the novelty of the results compared to the related work, e.g., mentioned in the comments, should be pointed out.


Some further questions and remarks:

(Q1) Did you use a validation set for evaluation? If not, what kind of stopping criteria did was use?

(Q2) You use the universal approximation theorem to prove Theorem 3. Could you please say something about the needed width of the networks?

(R1) Could you please provide standard deviations for all experiments. I suspect that the accuracies on the these small datasets fluctuates quite a bit.

(R2) In the comments it was already mentioned, that some important related work, e.g., [1], [2], are not mentioned. You should address how your work is different from theirs.


Minor remarks:

- The colors in Figure 1 are difficult to distinguish



[1] https://ieeexplore.ieee.org/stamp/stamp.jsp?tp=&arnumber=4703190
[2] https://people.csail.mit.edu/taolei/papers/icml17.pdf

-------------------
Update:
Most of the weak points were appropriately addressed by the authors and I have increased my rating accordingly.

---

> ### Author Response · Authors · 2018-11-22
> **Response to Reviewer2**
>
> Thank you for the detailed reviews. In the general post, we have addressed your chief concern regarding our original Eqn (4.1) and part of Theorem 3a). We sincerely hope R2 can revisit the rating in light of our revision and response.
>
> Answers to R2’s other questions:
> 1. RE: Standard deviations
> We added the standard deviations in Table 1. Note that on many datasets, standard deviations are fairly high for the previous methods as well as our methods due to the small training datasets. Our GINs achieved statistically significant improvement on the two REDDIT datasets where the number of graphs are fairly large. We leave the empirical evaluation on larger datasets to future work, but we believe that more expressive GNN models like our GINs can benefit more from larger training data by better capturing important discriminative structural features.
>
> 2. RE: Discussion on related work
> Following the suggestion, in Section 6 of the revised paper, we discuss the difference of our work to e.g., [1][2]. In short, the important difference is that  [1][2] both focus on the specific GNN architectures, while we provide a general framework for analyzing and characterizing the expressive power of a broad class of GNNs in the literature.
>
> 3. RE: Experimental setup and stopping criteria
> We selected an epoch with the highest cross-validation accuracy (averaged over 10 folds) following what previous deep learning papers do, e.g., [3][4]. This is for fair comparison as most previous papers on graph classification only report cross-validation accuracy.
>
> 4. RE: Network width
> Our proofs focus on existential analysis, i.e., there exists a way we can represent multisets with unique representations. Thus, the network width necessary for the functions provided in our proofs may only serve as an upper bound. For practical purposes, in our experiments, we found 32 or 64 hidden units are usually sufficient to perfectly fit the training set.
>
> [3] Mathias Niepert, Mohamed Ahmed, and Konstantin Kutzkov. Learning convolutional neural networks for graphs. In International Conference on Machine Learning (ICML), pp. 2014–2023, 2016.
> [4] Sergey Ivanov and Evgeny Burnaev. Anonymous walk embeddings. In International Conference on Machine Learning (ICML), pp. 2191–2200, 2018.

---

> > ### Comment · AnonReviewer2 · 2018-11-29
> > **Experimental setup**
> >
> > Thanks for your detailed reply. The mentioned weak points 1, 2 and 4 were appropriately addressed by the authors and I have increased my rating accordingly.
> >
> > Regarding point 3.
> > >> We selected an epoch with the highest cross-validation accuracy (averaged over 10 folds) following what previous deep learning papers do, e.g., [3][4].
> >
> > I think there is no common approach to this and the experimental setup in previous papers differs. Many papers use nested cross-validation, others use cross-validation with a fixed validation set, e.g., [5]. Also in [4] a validation seems to be used.
> > If I understand your method correctly, you report the best accuracy value obtained for any combination of hyperparameters -- instead of applying the classifier with the hyperparameters that work best for a validation set to the test set. In my opinion the approach is problematic. In particular, comparing to accuracy results obtained with a different experimental setup is not meaningful.
> >
> > [5] Hierarchical Graph Representation Learning with Differentiable Pooling
> > Rex Ying, Jiaxuan You, Christopher Morris, Xiang Ren, William L. Hamilton, Jure Leskovec
> > NeurIPS 2019

---

> > > ### Author Response · Authors · 2018-11-30
> > > **Response to R2**
> > >
> > > Thank you for the response. We address your question regarding experimental setup. First, past work in graph classification report the best cross-validation accuracy as what we did in our experiments [3]. The graph classification dataset sizes are often small, and therefore using a (single) validation dataset to select hyper-parameters is very unstable (for instance, MUTAG only has 180 data points, so each validation set only contains 18 data points. Compare this to standard deep learning benchmark sets like MNIST that has 70000 data points)**. Therefore, in our paper, we reported cross-validation accuracy for fair comparison to the previous methods. Moreover, our GNN variants and the WL kernel all follow the same experimental setups, so the comparison among them is definitely meaningful; consequently, our conclusion regarding the expressive power is also meaningful. We are planning for future work to evaluate our method on larger datasets, e.g. those mentioned in the post by one of our readers Mr. Christopher Morris, in https://openreview.net/forum?id=ryGs6iA5Km&noteId=B1xLcPaKpQ.
> > >
> > > We have thoroughly addressed all the concerns of R2. If Reviewer2 still has other questions or concerns regarding our work, we are happy to answer them.
> > >
> > > **[5] uses a test set, but its experiments focus on the larger datasets.

---

> > > > ### Comment · AnonReviewer2 · 2018-11-30
> > > > **Experimental setup, part 2**
> > > >
> > > > Even though the same approach was used in a previous paper, it is not convincing. Typically the results vary greatly between the epochs. Picking the one with the best validation accuracy leads to unrealistic results. Also the comparison to the results of the WL kernel is not meaningful since it was obtained with an SVM, where the number of hyperparameters is less. Therefore, you cannot pick the best value from such a large set of values. It is questionable to speak of "generalization" in the discussion of your results.
> > > >
> > > > I would like to propose to state the method you used more clearly in the paper and check the experimental setup used to obtain the results you have copied from other papers.
> > > >
> > > > Since the main contribution of the paper is theoretical, I will keep my rating, although I think that the experimental setup is a clear weak point.

---

> > > > > ### Author Response · Authors · 2018-11-30
> > > > > **Response to R2, Part II**
> > > > >
> > > > > Regarding experimental setup, we emphasize again that the graph classification datasets are extremely small compared to standard benchmarks in computer vision and NLP, e.g. ImageNet. Therefore, using a (single) validation dataset to select hyper-parameters is very unstable (for instance, MUTAG only has 180 data points, so each validation set only contains 18 data points). Therefore, following some of the previous deep learning work, we reported the ordinary cross-validation accuracy (the same hyper-parameters, such as number of epochs and minibatch size, were used for the entire folds). That being said, we understand the existing benchmarks and evaluation for graph classification are limited and we should all move on to large datasets as an anonymous reader pointed out in https://openreview.net/forum?id=ryGs6iA5Km&noteId=ryGs6iA5Km&noteId=H1gkUYX76Q. In the final version, we will also state our experimental setup more clearly. Thank you for your nice suggestion.

---

### Official Review · AnonReviewer1 · 2018-11-04
**One of the better GNN papers; would improve a lot with more careful discussion/analysis**

**Rating:** 7
**Confidence:** 5

**Review:**

This papers presents an interesting take on Weisfeiler-Lehman-type GNNs, where it shows that a WL-GNNs classification power is related to its ability to represent multisets. The authors show a few exemplar networks where the mean and the max aggregators are unable to distinguish different multisets, thus losing classification power. The paper also proposes averaging the node representation with its neighbors (foregoing the “concatenate” function) and using sum pooling rather than mean pooling as aggregator. All these observations are wrapped up in a GNN, called GIN. The experiments on Table 1 are inconclusive, unfortunately, as the average accuracies of the different methods are often close and there are no confidence intervals and statistical tests to help guide the reader to understand the significance of the results.

My chief concern is equating the Weisfeiler-Lehman test (WL-test) with Weisfeiler-Lehman-type GNNs (WL-GNNs). The WL-test relies on countable set inputs and injective hash functions. Here, the paper is oversimplifying the WL-GNN problem. After the first layer, a WL-GNN is operating on uncountable sets. On uncountable sets, saying that a function is injective does not tells us much about it; we need a measure of how closely packed we find the points in the function’s image (a measure in measure theory, a density in probability). On countable sets, saying a function is injective tells us much about the function. Moreover, the WL-test hash function does not even need to operate over sets with total or even partial orders. As a neural network, the WL-GNN “hash” ($f$ in the paper) must operate over a totally ordered set (\mathbb{R}^n, n > 0). Porting the WL-test argument of “convergence to unique isomorphic fingerprints” to a WL-GNN requires a measure-theoretic analysis of the output of the WL-GNN layers, and careful analysis if the total order of the set does not create attractors when they are applied recursively.

To illustrate the above *attractor* point, let’s consider the construct of Theorem 1 of (Xu et al., 2018), where the WL-GNN “hash” ($f$) is (roughly) described as the transition probability matrix of a random walk on the input graph. Under well-known conditions, the successive application of this operator ("hash" or transition probability matrix P in this case) can go towards an attractor (the steady state). Here, we need a measure-theoretic analysis of the “hash” even if it is bijective: random walk mixing. The random walk transition operator can be invertible (bijective), but we still say the random walker will mix, i.e., the walker forgets where it started, even if the transition operation can be perfectly undone by inversion (P^{-1}). In a WL-GNN that only uses the last layer for classification, this would manifest itself as poor performance in a WL-GNN with a large number of layers, and vanishing gradients. Of course, since (Xu et al., 2018) argued to revert back to the framework of (Duvenaud et al., 2015) of using the embeddings of all layers, one can argue that this mixing problem is just a problem of “wasted computation”.

The matrix analysis of the last paragraph also points to another potential problem with the sum aggregator. GIN needs to be shallow. With ReLU activations the reason is simple: for an adjacency matrix $A$, the value of $A^j$ grows very quickly with $j$ (diverges). With sigmoid activations, GIN would experience vanishing gradients in graphs with high variance in node degrees.

The paper should be careful with oversimplifications. Simplifications are useful for insight but can be dangerous if not prefaced by clear warnings and a good understanding of their limitations. I am not asking for a measure-theoretic analysis revision of the paper (it could be left to a follow-up paper). I am asking for a *relatively long* discussion of the limitations of the analysis.

Suggestions to strengthen the paper:
•	Please address the above concerns.
•	Table 1 should have confidence intervals (a statistical analysis of significance would be a welcome bonus).
•	Please mention the classes of graphs where the WL-test cannot distinguish two non-isomorphic graphs. See (Douglas, 2011), (Cai et al., 1992) and (Evdokimov and Ponomarenko, 1999) for the examples. It is important for the WL-GNN literature to keep track of the more fundamental limitations of the method.
•	(Hamilton et al, 2017) also uses the LSTM aggregator, besides max aggregator and mean aggregator, which outperforms both max and mean in some tasks. Does the LSTM aggregator also outperforms the sum aggregator in the tasks of Table 1? It is important for the community to know if unusual aggregators (such as the asymmetric LSTM) have some yet-to-be-discovered class-distinguishing power.


--------- Update -------

The counter-example in
https://openreview.net/forum?id=ryGs6iA5Km&noteId=rkl2Q1Qi6X
is indeed a problem for Theorem 3 if  \{h_v^{(k-1)}, h_u^{(k-1)} : u \in \mathcal{N}_v\} is not a typo for a set of tuples \{(h_v^{(k-1)}, h_u^{(k-1)}) : u \in \mathcal{N}_v\}. Unfortunately, in their proof, the submission states "difficulty in proving this form of aggregation mainly lies in the fact that it does not immediately distinguish the root or central node from its neighbors", which means \{h_v^{(k-1)}, h_u^{(k-1)} : u \in \mathcal{N}_v\} is actually \{h_v^{(k-1)}\} \cup \{ h_u^{(k-1)} : u \in \mathcal{N}_v\}, which is not as powerful as WL. Concatenating is more powerful than the summing the node's own embedding, but it results in a  simpler model and could be easier to learn in practice. And I am still concerned about the countable x uncountable domain/image issue I raised in my review.

Still, the reviewers seem to be doing all the discussion among themselves, with no input from the authors. I am now following Reviewer 2.

----

Reverting my score to my original score. The authors have addressed most of my concerns, thank you. The restricted theorems and propositions better describe the contribution.

I would like to note that while the proof of (Xu et al., 2018) is limited that does not mean it is not applicable to GIN or GraphSAGE or similar models. The paper uses 5 GNN layers, which in my experience is the maximum I could ever use with GNNs without seeing a degradation in performance. I don't think this should be a topic for this paper, though.


Xu, K., Li, C., Tian, Y., Sonobe, T., Kawarabayashi, K., & Jegelka, S. (2018). Representation Learning on Graphs with Jumping Knowledge Networks. In ICML.

Cai, J. Y., Fürer, M., & Immerman, N. (1992). An optimal lower bound on the number of variables for graph identification. Combinatorica, 12(4), 389-410.

Douglas, B. L. (2011). The Weisfeiler-Lehman method and graph isomorphism testing. arXiv preprint arXiv:1101.5211.

Evdokimov, S., & Ponomarenko, I. (1999). Isomorphism of coloured graphs with slowly increasing multiplicity of Jordan blocks. Combinatorica, 19(3), 321-333.

---

> ### Comment · AnonReviewer2 · 2018-11-22
> **The counterexample applies to Theorem 3**
>
> Theorem 3 states (as a sidline) that it makes no difference whether we consider a) (label(v), {label(u) : uv in E}) or b) just the set {label(v)} \cup  {label(u) : uv in E}. The set notation used for b) in the paper is a bit unclear, but this appears to be the intended meaning (from the proof and the approach used in section 4.1). For this set, Equation (4.1) yields an injection as claimed. Therefore the error actually affects Theorem 3, the main result of the paper. Clearly, WL is not perfect (otherwise it would solve the graph isomorphism problem), but that does not make the flaw any less serious. In my opinion, a revision of the authors is absolutely necessary.

---

> > ### Author Response · Authors · 2018-11-22
> > **The concern will be addressed soon.**
> >
> > We are now working hard for the thorough response and revision to fully address the concern of Reviewer2 and the anonymous reader. Thanks for your patience.

---

> > ### Comment · AnonReviewer1 · 2018-11-22
> > **Indeed, Theorem 3 is problematic (and the notation is confusing)**
> >
> > I understood \{h_v^{(k-1)}, h_u^{(k-1)} : u \in \mathcal{N}_v\} as a typo for a set of tuples \{(h_v^{(k-1)}, h_u^{(k-1)}) : u \in \mathcal{N}_v\}.  Which would have been fine.
> >
> > But you are right that looking at the proof in the appendix, it states "difficulty in proving this form of aggregation mainly lies in the fact that it does not immediately distinguish the root or central node from its neighbors" ... which is not how WL is supposed to work. Thanks!
> >
> > On top of these issues, WL requires a countable space while their approach operates over uncountable spaces (which remains my main concern). Even reverting to aggregation will not fix this mismatch.

---

> ### Author Response · Authors · 2018-11-22
> **Response to Reviewer1**
>
> Thank you for the detailed reviews and constructive feedback! We are glad that the reviewer finds our paper interesting. We apologize for the somewhat delayed response; it took us time to run additional experiments and add more careful analysis so that we can present an improved and more polished paper to everyone. We appreciate your understanding.
>
> In the following, we first address the main concern on equating the WL test and the WL-GNNs by showing its validity under a mild practical assumption. Then, we clarify the misunderstanding regarding the random walk mixing behavior of the WL-GNNs, showing that our GIN architecture does not suffer from such behavior. Finally, we discuss confidence intervals of our experimental results and also address other concerns of the reviewer.
>
> 1. RE: Validity of equating the WL test operating on countable sets to the WL-GNN operating on uncountable sets.
> The reviewer makes a great observation that countability of node features is essential and necessary for our theory, and we acknowledge that our current Theorem 3 and Lemma 5 are built on the common assumption that input node features are from a countable universe. We have now made this clear in our paper. We also filled in a technical gap/detail to address R1’s concern that after the first iteration, we are in an uncountable universe: this actually does not happen. We can show that for a fixed aggregation function, hidden node features also form a countable universe, because the countability of input node features recursively propagates into deeper layers. We also added a rigorous proof for this (Lemma 4 in our revised paper). As the reviewer nicely suggests, for the uncountable setting, it would be useful to have measure-theoretic analysis, which we leave for future work. Often input node features in graph classification applications (e.g., chemistry, bioinformatics, social) come from a countable (in fact, finite) universe, so our assumption is realistic. In the revised version, we clearly stated our assumptions at the beginning of Section 3 and have added further discussion on the relation between the WL test and WL-GNN after Theorem 3.
>
> 2. RE: Random walk mixing behavior of the GIN architecture.
> We think there might be a slight misunderstanding here: (1) Theorem 1 of (Xu et al., 2018) relates the random walk to the influence distribution in Definition 3.1 of (Xu et al., 2018), rather than the precise node representation, and (2) the analysis of Theorem 1 is specific to the GCN architecture (Kipf & Welling, 2017), where 1-layer perceptrons with mean pooling are used for neighbor aggregation. The GIN architecture does not suffer from the problem of random walk mixing because (1) Theorem 1 in (Xu et al., 2018) shows the influence distribution converges to a random walk limit distribution, however, it does not yet tell whether the node representations converge to the random walk limit distribution. Thus, “the walker forgetting where it started” may not happen.  (2) The GIN architecture uses MLPs rather than the 1-layer perceptron in (Kipf & Welling, 2017). The analysis in (Xu et al., 2018) specifically applies to models using 1-layer perceptrons, and therefore, it is not clear whether this analysis still holds for GIN.
> Furthermore, the reviewer is concerned with a possibly exploding value due to the sum aggregation, but this can be avoided because we have different learnable neural networks at each layer that can scale down the summed output (also, in practice, we did not observe such explosion).
>
> 3. RE: Confidence interval in experiments
> Following the suggestion, we added the standard deviations in Table 1. Because of space limit, we only added standard deviation in Table1, and confidence interval can be obtained via the standard deviation. The confidence interval of 95% is mean 0.754*std, and confidence interval of 90% is mean 0.611*std. Note that on many datasets, standard deviations are fairly high for the previous methods as well as our methods due to the small training datasets. Our GINs achieved statistically significant improvement on the two REDDIT datasets where the number of graphs are fairly large. We leave the empirical evaluation on larger datasets to future work, but we believe that more expressive GNN models like our GINs can benefit more from larger training data by better capturing important discriminative structural features.
>
> 4. Other comments:
> We also thank the reviewer for many other comments to strengthen our paper. In the revised paper, we clarified that WL-test cannot distinguish e.g., regular graphs. We discussed in Section 5.5 that the expressive power of other poolings such as LSTM and attention pooling can be analyzed under our framework, but we leave the empirical investigation to future work.

---

> > ### Comment · AnonReviewer1 · 2018-11-29
> > **Further corrections**
> >
> > For 10-fold cross validation, it is important to highlight that it tends to underestimate the confidence interval range (see (Bengio and Grandvalet, 2004)). Important to let readers know that there is more uncertainty in the results, which was not quantified.
> >
> > I also find the use of boldface confusing. Summing and subtracting the confidence intervals and a lot more models overlap.
> >
> > Bengio, Yoshua, and Yves Grandvalet. "No unbiased estimator of the variance of k-fold cross-validation." Journal of machine learning research 5, no. Sep (2004): 1089-1105.

---

> > > ### Author Response · Authors · 2018-11-30
> > > **Response to R1 (updated)**
> > >
> > > Thank you for the detailed response. Regarding the depth of the networks, GIN does not suffer from the curse of depth, i.e. we can use many layers, because we apply architectures similar to JK-Nets (specifically, JK-Concat) in Xu et al. 2018 for readout as described in Section 4.2. We conducted graph classification experiments using 5-layers GNNs (with JK-net) and they work nicely in our experiments. Moreover, as R1 nicely suggested, the influence distribution expansion phenomenon in Xu et al. 2018 indeed would apply to GraphSAGE, GIN etc, though the transition probabilities may not follow canonical random walks when MLP is applied. That being said, Xu et al. 2018 is a great work and we like it. We just wanted to clarify that Theorem 1 was about influence distribution rather than node features, thus, there would be no issue for GIN in terms of invertibility. We hope you are happy with our clarification.
> > >
> > > Regarding cross-validation, thanks for letting us know the work. We will mention it in the final version. To clarify, we use the boldface to indicate the best performance in terms of mean accuracy. As we mentioned in the rebuttal, the graph classification benchmark datasets are extremely small compared to other standard deep learning benchmarks for computer vision or NLP, e.g. ImageNet. That’s why standard deviations are high for all the methods (including the previous methods). We do believe we all should move beyond the conventional evaluation on these small datasets, but that is beyond the scope of this paper.
> > >
> > > Thank you again for your nice suggestions and detailed reviews. We hope our clarification regarding the analysis addresses your concerns.

---

> > > ### Author Response · Authors · 2018-12-04
> > > **Response to R1 (updated)**
> > >
> > > Thank you for the detailed response. Regarding the depth of the networks, GIN does not suffer from the curse of depth, i.e. we can use many layers, because we apply architectures similar to JK-Nets (specifically, JK-Concat) in Xu et al. 2018 for readout as described in Section 4.2. We conducted graph classification experiments using 5-layers GNNs (with JK-net) and they work nicely in our experiments. Moreover, as R1 nicely suggested, the influence distribution expansion phenomenon in Xu et al. 2018 indeed would apply to GraphSAGE, GIN etc, though the transition probabilities may not follow canonical random walks when MLP is applied. That being said, Xu et al. 2018 is a great work and we like it. We just wanted to clarify that Theorem 1 was about influence distribution rather than node features, thus, there would be no issue for GIN in terms of invertibility. We hope you are happy with our clarification.
> > >
> > > Regarding cross-validation, thanks for letting us know the work. We will mention it in the final version. To clarify, we use the boldface to indicate the best performance in terms of mean accuracy. As we mentioned in the rebuttal, the graph classification benchmark datasets are extremely small compared to other standard deep learning benchmarks for computer vision or NLP, e.g. ImageNet. That’s why standard deviations are high for all the methods (including the previous methods). We do believe we all should move beyond the conventional evaluation on these small datasets, but that is beyond the scope of this paper.
> > >
> > > Thank you again for your nice suggestions and detailed reviews. We hope our clarification regarding the analysis addresses your concerns.

---

### Public Comment · (anonymous) · 2018-10-07
**So why GIN still outperforms WL kernel on some dataset?**

Since the GIN is developed to achieve as strong expressive power as WL graph isomorphism test, why does it still has much better result on reddit-binary and reddit-5K than WL subtree Kernel? Do you also tried on larger dataset such as reddit-12K?

---

> ### Author Response · Authors · 2018-10-07
> **Because GIN can capture similarity between different subtrees.**
>
> Thanks for your questions!
>
> As we mentioned in Section 4 right after Theorem 3, GIN generalizes the WL graph isomorphism test by learning to embed the subtrees to continuous space. This enables GIN to not only discriminate different structures, but also to learn to map similar graph structures to similar embeddings and capture dependencies between graph structures. Such learned embeddings are particularly helpful for generalization when the co-occurrence of subtrees is sparse across different graphs or there are noisy edges (Yanardag & Vishwanathan, 2015).
>
> Regarding the dataset, we did not try reddit-12K at this moment.

---

### Public Comment · (anonymous) · 2018-10-08
**Remarks regarding the use of ReLU as non-linearity**

Thank you for this very interesting work which gives a lot of insight into graph neural networks and structures the large amount of related work out there.

I have some remarks/opinions regarding the use of non-linearities in this work.

1) Regarding section 5.1 and lemma 5: I do not think that more than 1 layer is necessary. The ReLU non-linearity does only show its full potential when used together with a bias. In most literature, the bias term is (unfortunately) omitted in the paper but still used in the implementation. ReLU without bias separates based on a hyperplane which always goes through the origin, which is why the example in the proof of Lemma 5 works. All values lie in one piece-wise linear subspace of the functions range. When using a bias, the non-linear point can be shifted to separate both examples in a non-linear fashion and the example that proves Lemma 5 does not work anymore. I am not sure though if there is another example that works if a bias is present. I suspect though, that one layer with a "working non-linearity", e.g. ReLU with bias, should be enough.

Therefore, I guess the insight here is: We need a (working) non-linear mapping before doing the feature aggregation (assuming no one-hot encoding), otherwise, we lose injectivity and therefore, discriminative power. In many current GNN models (including GCNs), this is not the case.

2) Further, I suspect that depending on how the COMBINE operation is defined, the discriminative power of WL can also be obtained by stacking 2 layers in the following way:
Assuming COMBINE to be \sigma ( W_1*x + W_2*y + b), with x being the result of neighbourhood aggregation and y the last current node feature. Further, in the first layer, let the features from the neighbourhood aggregation get discarded (W_1 = 0), resulting in a node-wise fully connected layer with nonlinearity (or "1x1-convolution" or however it might be called).
Then, the second layer receives features which went through a non-linear function before aggregation. Since the network could learn W_1 = 0, those two layers should have the same discriminative power as the WL.

3) I think the formulation of GCN in Equation 2.2 is not correct. The original GCN aggregates first and applies the non-linearity afterwards.
It should be noted that since GCN does not have individual W's for the root node and the neighbourhood (W_1 and W_2 in the equation above) the mentioned construction from 2) does not work here.

---

> ### Author Response · Authors · 2018-10-09
> **Thanks for the discussion!**
>
> Thank you for your interest and positive comments on our work! Let us try to answer your questions. There are many GNN formulations. So it is always interesting to understand the power of different variants!
>
> 1) Thanks for this insightful comment! With sufficiently large dimensionality of output units, ReLU with bias might indeed be able to distinguish different multisets (larger output dimensionality is generally needed as we have more multisets to distinguish). In our experiments, we actually had the bias term, and we empirically observed that under-fitting still sometimes occurred for models with 1-layer perceptrons (with bias) (see Figure 4). We think it could be due to the limited number of output units or optimization.
>
> We would like to emphasize that with MLPs, we can enjoy universal approximation of multiset functions. This allows Sum-MLP (GIN) to go beyond just distinguishing different multisets and to learn suitable representations that are useful for applications of interest. In fact, Sum-MLP outperformed Sum-1-layer in 7 out of 9 datasets (comparable in the other 2) in terms of test accuracy!
>
> We will further discuss these points and practical implications in our updated version.
>
> 2) There can certainly be other GNN architectures with the same discriminative power as GIN (as long as they satisfy conditions in our Theorem 3). Your proposed formulation with COMBINE could potentially also work, although we do not fully understand your description. It would be great future work to investigate other powerful GNN models with potentially better generalization and optimization.
>
> 3) (2.2) is indeed not exactly the same as the original GCN. Our emphasis here was that MEAN aggregation was used in GCN. We used the formulation (2.2) to share the same framework with GraphSAGE (MAX aggregation) to save space. We will include the exact formulation of GCN in the updated version. Also, we mentioned after (2.2) that GCN does not have a COMBINE step and aggregates a node along with its neighbors.

---

> > ### Public Comment · (anonymous) · 2018-10-09
> > **Clarification**
> >
> > Thank you for the discussion!
> >
> > I'd like to clarify my point 2) further (it is an observation, not criticism):
> >
> > Assuming we have a COMBINE operation like described above:   \sigma ( W_1*x + W_2*y + b)
> > If we now stack n layers (NO weight sharing over time) and assume W_1 = 0 for the first n-1 of them, we arrive exactly at the formulation where we have an MLP with n-1 layers, followed by a normal GNN layer.
> >
> > The point I wanted to make: There are architectures in current literature that already achieve injectivity (maybe by "accident") through this construction. Maybe it can be said: As long as there is an individual W for the self-connection, the condition can be fulfilled through stacking.
> >
> > Examples are:
> > Defferrard et al.: Convolutional Neural Networks on Graphs with Fast Localized Spectral Filtering, 2016 (individual parameter for k=0 neighbourhood)
> > Gilmer et al.: Neural Message Passing for Quantum Chemistry, 2017 (depending on implementation, i guess)

---

> > > ### Author Response · Authors · 2018-10-09
> > > **We develop theory to turn “by accident” into “common practice”**
> > >
> > > That’s a good observation. Indeed, there are great stuff in the nature possibly found by accident, e.g. rare grasses in Chinese medicine. Here, our goal is to study and develop theory to understand the underlying principles, so that we can appreciate the great stuff, and that in the future, with the insight of our theory, we can build even better graph deep learning models!

---

### Public Comment · (anonymous) · 2018-10-10
**regarding lemma 4**

The proof of Lemma 4 assumes the graphs have a constant degree bound (|X|<N). Is the statement true even in general (i.e., finite |X|, but not bounded by a constant)? E.g., in inductive setting test graphs could have high degree.

---

> ### Author Response · Authors · 2018-10-10
> **The node degrees can be arbitrarily large**
>
> Thank you for your interest! The finite node degrees |X| can be arbitrarily large, and we can always find an N that works (we do not have to put an upper bound on N). Note that Lemma 4 only shows the existence of injective functions, and in practice, we need our neural networks to learn these functions from data.

---

> > ### Public Comment · (anonymous) · 2018-10-11
> > **clarification**
> >
> > Could you clarify how you can always find an N that works without an upper bound? My understanding is that N should be at least as large as the largest degree you would encounter in the set of all training + testing graphs, for the function to be injective in all of these graphs. Please correct me if I am wrong.
> >
> > If the set of training + testing graphs are bounded in size, sure I can pick a large constant for N and that should work. But it's possible the distribution of graphs includes graphs of unbounded size (e.g., number of nodes drawn from a geometric distribution). What N should I pick then?
> >
> > In practice, of course, all graphs have bounded size and it doesn't matter. But I want to understand what is the precise theoretical statement to be made here.

---

> > > ### Author Response · Authors · 2018-10-11
> > > **Reply**
> > >
> > > You are right; we can simply pick sufficiently large N that is bigger than the size of any graphs of interest. Also, all graphs of our interest are of bounded sizes, and we explicitly stated in our Lemma 4 that we dealt with finite multiset*; thus, your second question does not make sense to us.
> > >
> > > *https://en.m.wikipedia.org/wiki/Finite_set

---

> > > > ### Public Comment · (anonymous) · 2019-01-06
> > > > **The sizes of all finite multisets cannot be bounded**
> > > >
> > > > In appendix D and E you state and use the fact that "because multisets X are finite, there exists a number N s.t. |X| < N for all (finite) X". This is mathematically incorrect, since arbitrarily large finite sets cannot have an upper bound in size. In fact, the set of all finite sets containing elements in a countable set is uncountable. The authors might want to clearly state the assumption that they deal with finite sets of cardinality at most N.

---

> > > > > ### Author Response · Authors · 2019-01-06
> > > > > **Thanks!**
> > > > >
> > > > > We appreciate your great suggestion, and we will clarify our assumption accordingly.

---

> > > > > > ### Public Comment · ~Octavian_Eugen_Ganea1 · 2019-01-07
> > > > > > **this assumption in lemma 5 can be relaxed**
> > > > > >
> > > > > > I believe you can restate lemma 5 to hold for any finite or infinite (but countable) multiset  with the assumption that each element x appears at most N - 1 times in this multiset. Then, the same function f(x) = N^{-Z(x)} can be used, and the convergent series \sum_{x \in X} f(x) would always be injective. This can be proven by induction and it reduces to the following case: if we have 2 series: S = \sum_{i >= 0} x_i * N^{-i} and T=\sum_{i >= 0} y_i * N^{-i} , with N-1 > x_i,y_i >=0, then, if x_0 < y_0, one can prove that S < T.

---

> > > > > > > ### Author Response · Authors · 2019-01-07
> > > > > > > **Thanks everyone!**
> > > > > > >
> > > > > > > Thank you everyone for reading and liking our paper, as well as giving many good suggestions. Happy holidays!

---

### Public Comment · (anonymous) · 2018-10-12
**Related work, difference to older work**

There is an article from 2009 [1] which has a similar theoretical contribution. Could you please comment on the differences.

[1] https://ieeexplore.ieee.org/stamp/stamp.jsp?tp=&arnumber=4703190

---

> ### Public Comment · (anonymous) · 2018-10-13
> **Connections**
>
> I am also curious if there is any connections here. From my understanding, one difference is that Scarselli et al. (2009) focus on a specific type of GNN (with a recurrent contraction aggregator), so the analysis probably doesn't apply to mordern GNN architectures like GCN.  On the other hand, this paper provides a general framework that gives insight to a number of GNN architectures.

---

> > ### Author Response · Authors · 2018-10-13
> > **We provide a general framework for analyzing and rethinking a large amount of Graph NNs in the literature.**
> >
> > We thank both Anonymous 1 and Anonymous 2 for your interest in our work!
> >
> > To Anonymous 1:  Thanks for bringing up this early work! We will comment on the differences below. We would like to refer to Anonymous 2’s comment first, which made a very good point.
> >
> > To Anonymous 2: Thank you for the insightful comment! Indeed, [1] analyzes a specific model with recurrent contraction maps, but our analysis framework applies to general GNNs with message passing/neighbor aggregation. Regarding the connection and differences of contraction, recurrent maps and more general aggregators, the talk/paper by Yujia Li et al [3][4] provide some very good explanations and insights! Highly recommended!
> >
> > More detailed explanations on the differences:
> >
> > 1) As Anonymous 2 pointed out, the 2009 paper [1] analyzes a specific architecture designed in [2] that uses contraction maps and the same aggregator in all layers. Although [1] proves [2] can capture rooted subtree structures, it has been observed e.g. in [3][4], that it does not perform ideally in practice, thus leading to the surge of a large amount of modern GNN architectures like Gated GNN, GCN, GraphSAGE etc. Our architecture GIN is shown to perform well in practice. To Anonymous 2: in our preliminary experiments, we also tried sharing the same aggregator across all layers of GNN, but the training accuracy was fairly low (usually < 80%), possibly due to optimization or capacity issues.
> >
> > 2) While [1] focuses on the specific GNN in [2], we provide a general framework for characterizing the expressive power of many different GNN variants proposed so far in the literature. Our results are not only applicable to [2], GIN etc. but also applicable to almost all modern GNN architectures like GCNs and GraphSAGE.
> >
> > 3) We made an explicit comparison of different GNN variants both theoretically and empirically so that we can have better understandings of their theoretical properties. Specifically,  we characterized what graph substructures different aggregation schemes can capture, and discussed how that might affect empirical performance. We also made it clear that injectiveness of the aggregation function is the key to achieving high expressive power in GNNs.
> >
> > Therefore, we believe our work plays an important role in rethinking and structuring the 10-year literature of GNNs from the viewpoint of expressive power, despite some similarity to [1] in terms of capturing rooted subtree structures. We will also discuss [1] and [2] in our updated version.
> >
> >
> > [1] Scarselli, Franco, et al. "Computational capabilities of graph neural networks." IEEE Transactions on Neural Networks 20.1 (2009): 81-102.
> > [2] Scarselli, Franco, et al. "The graph neural network model." IEEE Transactions on Neural Networks 20.1 (2009): 61-80.
> > [3] https://www.cs.toronto.edu/~yujiali/files/talks/iclr16_ggnn_talk.pdf
> > [4] Li, Yujia, et al. "Gated graph sequence neural networks." arXiv preprint arXiv:1511.05493 (2015).

---

> > > ### Public Comment · ~Christopher_Morris1 · 2018-10-16
> > > **More important related work**
> > >
> > > I think you also miss other important related work [5], which shows that the features computed by GNNs lie in the same Hilbert space as WL.
> > >
> > >
> > > [5] https://people.csail.mit.edu/taolei/papers/icml17.pdf

---

> > > > ### Author Response · Authors · 2018-10-19
> > > > **Thank you everyone.**
> > > >
> > > > We thank everyone for interest and many inquiries about our work.
> > > >
> > > > To Anonymous 3: Thanks for bringing up this related work. Graph representation learning is an increasingly popular research topic with a surge of many wonderful works. We will make sure to add all the relevant references in our updated version. To emphasize the difference with the related work, [5] shows their proposed architecture lies in the RKHS of graph kernels, but does not tell anything about which graphs can actually be discriminated by the network. In contrast, we address the question of which graphs can be distinguished, and provide a framework for addressing this representational question in a general way, settling the representational power of a broad class of GNNs.

---

### Public Comment · (anonymous) · 2018-10-13
**Thoughtful and provocative work! Future directions?**

Thanks for the thoughtful and provocative work! The paper answered some questions I have been thinking about. Graph convolution that many people talk about was motivated by Fourier transform of graph Laplacian and analogy with computer vision, yet I thought it’s not quite the same as vision. I was curious what are the more natural explanations. The view of “capturing graph structures with powerful aggregators” sounds much more natural to me and also natural to graphs problems. Very provocative!

I wonder what possible good future directions look like for graphs? Many great works these years apply theoretical computer science techniques to machine learning, e.g. Prof Sanjeev Arora group from Princeton and Prof. Aleksander Madry group from MIT. Do you see similar directions for graphs?

---

> ### Author Response · Authors · 2018-10-13
> **Our thoughts.**
>
> Thank you for your interest in our work!
>
> Great that you found the framework presented in our paper intuitive/natural for understanding graph representations. We think the spectral perspectives [1] [2] also provide a very valuable and important angle. It would be interesting to understand how to connect and relate the different perspectives. Regarding future directions, besides what we have mentioned in our conclusion, we do not have further comments at this moment. Combining and applying techniques from many other communities indeed sounds very interesting and promising. Ideas from graph minor theory [3] and spectral graph theory [4] [5] may be interesting and are not fully explored in the current message passing frameworks, although we do not have detailed suggestions at the moment.
>
> [1] Bruna, J., Zaremba, W., Szlam, A., and LeCun, Y. Spectral networks and locally connected networks on graphs. International Conference on Learning Representations (ICLR), 2014.
> [2] Bronstein, M. Bruna, J., Szlam, A., LeCun, Y. and Vandergyst, P. Geometric Deep Learning: going beyond Euclidean Data IEEE Sig. Proc. Magazine, 2017
> [3] https://www.birs.ca/workshops/2008/08w5079/report08w5079.pdf
> [4] http://www.cs.yale.edu/homes/spielman/561/
> [5] http://courses.csail.mit.edu/6.S978/

---

> > ### Public Comment · (anonymous) · 2018-10-16
> > **Thank you so much for the reply!**
> >
> > Thank you so much for providing possible ideas for future directions! The materials you referenced look very helpful and I will take a look at graph minor theory and spectral graph theory.

---

### Public Comment · (anonymous) · 2018-10-19
**Some questions about the paper**

Hi！I'm writing to ask some questions.

1. In Section 3, you said that "Intuitively, the most powerful GNN maps two nodes to the same location only if they have identical subtrees structures with identical features on the corresponding nodes".  However, in my opinion, a powerful model should map nodes with different labels into different locations instead of features, since there may be some noise in features.

2. In the paper, you said that GIN is the most powerful model. But you only reported experimental results on graph classification. Have you validated the proposed model on node classification tasks? Based on my understanding, it's also important to consider the performance on node classification when judging the power of a GNN model?

3. Instead of Mean/Max aggregators in GCN and GraphSAGE, MLP is used as the aggregator in each layer. Have you compared the parameter complexity with other baselines?

Thank you!

---

> ### Author Response · Authors · 2018-10-22
> **Answers**
>
> Thanks for your interest. Answers to your inquiries:
>
> 1. Note that being powerful entails “being able to” map nodes with different subtrees to different representations. If a model is not capable of achieving this, then it’s intrinsically less powerful in distinguishing different graphs. In addition, to combat noise, we can simply regularize the mapping function to be locally smooth (e.g., by using Virtual Adversarial Training [1]). Nonetheless, in many graph classification applications including those in our experiments, the node features have specific meanings (e.g. an atom of certain types) and are not noisy.
>
> 2. Note that our paper focuses on expressive power of GNNs, and there are two main reasons why it is not very interesting for us to conduct node classification experiments to validate our claim.
> First, as we have emphasized in Section 5 and 5.3, in many node classification applications, node features are rich and diverse (e.g. bag-of-words representation of papers in a citation network), so GNN models like GCN and GraphSAGE are often already able to fit the training data well. Second, many node classification tasks assume limited training labels (semi-supervised learning scenario); thus, the inductive bias of GNNs also plays a key role in empirical performance. For example, as we discussed in Section 5.3, the statistical and distributional information of neighborhood features may provide a strong signal for many node classification tasks.
>
> Our GINs may potentially perform well on node classification tasks. However, due to our explanations above, the performance on node classification tasks are less directly explained by our theory of representational power, so we leave the experiments for future work. We believe our experiments on graph classification are sufficient and great for validating our theoretical claim on expressive power of GNNs.
>
> 3. We set the numbers of hidden units and output units of MLP to be same. So the parameter complexity of Sum-MLP is roughly two times as many as that of Sum-Linear. However, note that with more hidden units, the performance of models with 1-layer perceptrons usually decreases.
>
> [1] https://arxiv.org/abs/1704.03976

---

### Public Comment · (anonymous) · 2018-10-22
**The role of discriminative power for graph classification**

I understand that GIN provably has more discriminative power than other variants of GNN. But the ability to differentiate non-isomorphic graphs does not necessarily imply better graph classification accuracy, right? Would it be possible to strong discriminative power will backfire for the graph classification? After all, we don't need to solve graph isomorphism here.

---

> ### Author Response · Authors · 2018-10-22
> **More powerful GNNs can better capture discriminative substructures of graphs**
>
> As we have pointed out in the experiment section, although stronger discriminative power does not directly imply better generalization, it is reasonable to expect that models that can accurately capture graph structures of interest also perform well on test set. In particular, with many existing GNNs, the discriminative power may not be enough to capture graph substructures that are important for classifying graphs. Therefore, we believe strong discriminative power is generally advantageous for graph classification. In our experiments, we empirically demonstrated that our powerful GIN has better generalization as well as better fitting to training datasets compared to other GNN variants. GINs performed the best in general, and achieved state-of-the-art test accuracy. We leave further theoretical investigation of generalization to our future work.

---

### Public Comment · (anonymous) · 2018-11-09
**Dataset problem**

A comment on the dataset. I think current dataset is very limited for evaluating different graph learning algorithms. A new paper showed that using very simple degree statistics already can perform on par with the state-of-the-art graph neural networks and graph kernel. Imagenet Like dataset is strongly needed for evaluating different algorithms fairly.

Reference:
A simple yet effective baseline for non-attribute graph classification https://arxiv.org/abs/1811.03508

---

> ### Public Comment · ~Christopher_Morris1 · 2018-11-14
> **Dataset problem**
>
> There are already larger real-world datasets available, see e.g., [1].
>
> [1] http://moleculenet.ai/datasets-1

---

> > ### Public Comment · (anonymous) · 2018-11-19
> > **But we still don't have a dataset that contain many large networks**
> >
> > Thanks for pointing out the dataset. But I believe those datasets contain many small graphs. A dataset of many large graphs is still missing.

---

### Public Comment · ~Christopher_Morris1 · 2018-11-15
**Problem with Equation (4.1)**

I do not think that Equation (4.1) is as powerful as the 1-WL. Consider the two labeled graphs

r -- g
|    |
g -- r

and

r -- g
|    |
r -- g

with node color "g" and "r". Clearly, the 1-WL can distinguish between these two graphs. Howeover, when using (4.1) with an 1-hot encoding of the labels, both graphs will end up with the same two features. The set of node features will always be the same.

---

> ### Comment · AnonReviewer2 · 2018-11-22
> **Relation to Theorem 3**
>
> The counterexample appears to be related to a flaw in Theorem 3, see this comment: https://openreview.net/forum?id=ryGs6iA5Km&noteId=ByxsKEkV07
>
> In my opinion, a statement of the authors (and a revision) is absolutely necessary.

---

> > ### Author Response · Authors · 2018-11-22
> > **Concern is addressed above.**
> >
> > We thoroughly addressed the counter-example and the related concern in https://openreview.net/forum?id=ryGs6iA5Km&noteId=SyeZ3MU4AX
> > Furthermore, we revised our paper.

---

### Author Response · Authors · 2018-11-22
**Modification of GIN aggregation to address the concern (Part 2).**

We also conducted extensive experiments on the modified GIN architecture with Eq. (**), where we learn epsilon by gradient descent. We included the additional results in Section 7 of our revised paper. In terms of training accuracy, which is the main focus of our paper, we observed from our new Figure 4 (in the revised paper) that the modified GIN (we call it GIN-eps in our paper) gives the same results as our original GIN (GIN-0) does, showing no improvement on the training accuracy. This is because the original GIN already fits the training data very well, achieving nearly 100% training accuracy on almost all of our datasets. Consequently, the explicit learning of epsilon in the modified GIN (GIN-eps) does not help much. Interestingly, in terms of the test accuracy, we observed from Table 1 (in the revised paper) that for GIN-eps (modified GIN) there is a slight drop in test accuracy  (0.5% on average) compared to GIN-0 (original GIN). Since GIN-0 and GIN-eps showed almost no difference in training accuracy, both have sufficient discriminative power on this data, and the slight drop in test accuracy should be explained by generalization rather than expressiveness. We leave the investigation of the effectiveness of GIN-0 for future work. We want to emphasize that the pooling scheme (sum vs. average vs. max) and mapping scheme (MLP vs. linear) does affect the performance w.r.t. training accuracy, and consequently also affects the test accuracy. Thus, our main findings distinguishing the sum-MLP architecture from other aggregation schemes for maximally expressive GNNs is still valid.

As a final remark, as R1 nicely commented, instead of Eq. (**), a node and neighbors can be concatenated, rather than summed, to achieve the same power as the WL test. Interestingly, as R1 cleverly predicted, in our preliminary experiments, we found such concatenation was harder to train compared to our simple GINs (both GIN-0 and GIN-eps) and achieved lower test accuracy. We leave the extensive investigation and comparison to our future work.

We sincerely appreciate the reviewer and commenter for the great suggestions and insights, which enabled us to further strengthen our work and make our paper stronger. We hope our new version resolves the reviewers’ main concerns.

---

### Author Response · Authors · 2018-11-22
**Modification of GIN aggregation to address the concern (Part 1).**

We begin by acknowledging that Eqn (4.1) and Theorem 3a-Eqn.2) in our initial submission (which does not distinguish the center nodes from their neighbors) were indeed insufficient to be as powerful as the WL test. The example provided by the anonymous reader makes a great point about the corner case. That said, we agree that in order to realize the most powerful GNN, its aggregation scheme needs to distinguish the center node from its neighbors.

The good news is that we can resolve this corner case by making a very simple modification to our GIN aggregation scheme in Eq. (4.1) of the initial submission, so that the modified GIN can provably distinguish the root/center node from its neighbors during the aggregation. This implies that our modified GIN handles the counter-example raised by the anonymous reader, and, more importantly, we can prove that the modified GIN is as powerful as the WL test under the common assumption that the input node features are from a countable universe. In the following, we will explain these points in more detail.

First, we present a simple update to our current GIN aggregation scheme, and show that it now handles the counter-example provided by the anonymous reader. Our simple modification to the original GIN aggregation in Eq. (4.1) of the initial submission is:

h_v^{(k)} = MLP ( (1 + \epsilon) h_v^(k-1) + \sum_{u \in neighbor} h_u^(k-1) ), (**), Eq. (4.1) of the revised paper.

where \epsilon is a fixed or learnable scalar parameter. We will show that there exist infinitely many \epsilon where the modified GIN (as defined above) is as powerful as WL. Note that setting \epsilon = 0 reduces to our original GIN aggregation in Eq. (4.1) of the initial submission. Thus, the above equation (Eq. (**)) smoothly “extrapolates” the original GIN architecture, and with the epsilon term, the modified GIN can now distinguish the center node from its neighbors. Before moving to the formal proof, let us first illustrate how modified GIN handles the counter-example by the anonymous reader:

 R - R                 R - G
|      |     v.s.    |      |
 G - G                G - R

Assume we use the one-hot encodings for the input node features, i.e., R = [1, 0] and G = [0, 1]. After 1 iteration of aggregation defined by Eq. (**), our modified GIN obtains the following node representations (before applying MLP in (**)); thus, it successfully distinguishes the two graphs with small non-zero eps=\epsilon.

[2+eps, 1]  -- [2+eps, 1]                  [1+eps, 2] -- [2, 1+eps]
|                            |           vs.         |                            |
|                            |                         |                            |
[1, 2+eps] -- [1, 2+eps]                   [2, 1+eps] -- [1+eps, 2]

The key here is that with non-zero (small) eps, [2+eps, 1] and [2, 1+eps] are now different. In other words, adding \epsilon term in Eq. (**) enables the modified GIN to “identify” the center nodes and distinguish them from neighboring nodes.

With the intuition above, we now give a formal proof for the modified GIN architecture. We start with Lemma 5 (universal multiset functions) in our revised paper, and extend it to Corollary 6 in the revised paper that can distinguish center node from the neighboring nodes. Crucially, the function h(c, X) in Corollary 6 is now the injective mapping over the *pair* of a center node c and its neighbor multiset X. This implies that h(c, X) in Corollary 6 can distinguish center nodes from their neighboring nodes.

Corollary 6
Assume \mathcalcal{X} is countable. There exists a function f: \mathcal{X} → R^n so that for infinitely many choices of \epsilon, including all irrational numbers, h(c, X) \equiv (1 + \epsilon) f(c) + \sum_{x \in X} f(x) is unique for each pair (c, X), where c \in \mathcal{X}, and X \subset \mathcal{X} is a finite multiset.

---Proof sketch (details are provided in Appendix of the revised paper, see Proof of Corollary 6)
The proof builds on Lemma 5 that constructs the function f that maps each finite multiset uniquely to a rational scalar with N-digit-expansion representation. With the same choice of f from Lemma 5, the irrationality of \epsilon enables us to distinguish the center node representation c from any combination of multiset representation, which is always rational. That is, h(c,X) is unique for each unique pair (c,X).
----

Using h(c, X) for the aggregation, we can straightforwardly derive our modified GIN aggregation in Eq. (**) (similarly to the MLP-sharing-across-layer trick described after Lemma 5.) We included a detailed derivation in Section 4.1 of the revised paper.

---

### Author Response · Authors · 2018-11-22
**Paper update overview**

We sincerely appreciate all the reviews, they give positive and high-quality comments on our paper with a lot of constructive feedback. We also thank the many anonymous commenters for their interest and helpful discussion. In the revised paper, we did our best to address the concerns and suggestions to strengthen our paper. We sincerely hope reviewers revisit the rating in light of our revision and response. The following summarizes our revisions. Please see our rebuttal for the detailed discussion.

Major revisions:
1. An anonymous reader and Reviewer2 made a clever observation that our original GIN aggregation in Eq. (4.1) and Theorem 3a-Eqn.2) of the initial submission and cannot distinguish certain corner case graphs that the WL test can distinguish. We fixed this issue by 1) making a slight modification to GIN’s aggregation in Eq. (4.1), and 2) adding Corollary 6 to show Eqn. (4.1) in the revised paper is as powerful as WL, 3) removed Theorem 3a-Eqn.2). The modified GIN aggregation smoothly extrapolates the original one, avoids the corner case, and can be shown to be as powerful as the WL test. We conducted extensive experiments on the modified GIN to further validate our model. (see below https://openreview.net/forum?id=ryGs6iA5Km&noteId=ryGs6iA5Km&noteId=SyeZ3MU4AX for our detailed response.)

2. Based on the helpful comments of Reviewer1 on countability of node features, we have now made our setting much clearer: We clarified the common assumption that input node features are from a countable set, and we further added Lemma 4 in the revised paper to prove that the hidden node features are also always from a countable set under this assumption. With the countability assumption, it is meaningful to discuss injectiveness in Theorem 3, and our countability assumption used in Lemma 5 (universal multiset functions) always holds. We also provided detailed discussion on the correspondence between the WL test and WL-GNN under the countability assumption, validating our theory to equate those two.


Minor revisions:
1. R3 makes a great point that beyond distinguishing different graphs, it is equally important for GNNs to capture their structural similarity. We have already mentioned this point after Theorem 3. We now made this clearer and added a more detailed discussion in Section 4.
2. In response to R3 and R2, we added Section 6 for detailed discussion of related work.
3. Following the suggestions by R1 and R2, we added standard deviations in the experiments.
4. Based on the great insight by an anonymous reader, we added discussion on the expressive power of Sum-Linear when the bias term is included.

---

### Public Comment · (anonymous) · 2018-11-28
**GIN is essentially the same as structure2vec?**

GIN is essentially the same as the graph neural network in
equation (10) of this paper:
Dai et al. ICML 2016. Discriminative Embeddings of Latent Variable Models for Structured Data
https://arxiv.org/pdf/1603.05629.pdf

A discussion of this related work, and compare to structure2vec in their datasets will help improve the paper.

Also how about the other message passing version of graph neural network developed in Dai et al. (eq (14) & (15)) ? Will it be more powerful?

---

> ### Public Comment · (anonymous) · 2018-11-28
> **All the graph Laplacian normalizations in previous GCN are not essential?**
>
> According to the current paper, can one say that all the graph Laplacian normalizations in previous GCN are not essential? Or redundant in some sense?
> What's really essentially in graph neural network is equation (4.1) for GIN, or equation (10) for structure2vec in Dai et al.?
> And a potentially really different representation power will probably come from a different message passing update as in eq (14) & (15) in Dai et al.?

---

> > ### Author Response · Authors · 2018-11-28
> > **No. GIN is different.**
> >
> > Thanks for your interest. GIN is different from the paper you mentioned. Critically, GIN uses MLP while Dai et al. uses perceptron.
> > There are many GNN variants and we leave the analysis of some of them for the future work. Note that the graph Laplacian normalization can decrease the representational power of GNNs, but it can also induce useful inductive bias for the applications of interest, e.g, semi-supervised learning. Therefore, we can not draw a decisive conclusion about the normalization only from the perspective of representational power. It is our future work to investigate generalization, inductive bias and optimization of different GNN variants.

---

> > > ### Public Comment · (anonymous) · 2018-11-30
> > > **You mean an MLP parameterization like equation (2) of this paper?**
> > >
> > > Neural network-based graph embedding for cross-platform binary code similarity detection
> > > https://arxiv.org/pdf/1708.06525.pdf

---

### Public Comment · ~Yangliao_Geng1 · 2023-11-28
**about the mean operation in Eq. (2.3) for Graph Convolutional Networks (GCN)**

Thanks for the excellent work. I have a little doubt about the mean operation of GCN in Eq. (2.3). It seems somewhat different from that in the original paper (Kipf & Welling, 2017). In (Kipf & Welling, 2017), since symmetric normalization is used for the adjacency matrix (or Laplacian matrix), it is actually a weighted average of the node features of a node's neighborhood (different nodes may have different weights). Therefore, using the mean operation to abstract seems unreasonable since the mean operation has permutation invariance, but the weighted average does not in general.

Furthermore, the proof in Lemma 2 ("The same input, i.e. neighborhood features, generates the same output") seems to rely on the permutation invariance of the mean operation. If a weighted average is used, will it affect the correctness of Lemma 2's proof?


Kipf, T. N., & Welling, M. (2016, November). Semi-Supervised Classification with Graph Convolutional Networks. In International Conference on Learning Representations.

---

### Meta-Review · Area_Chair1 · 2018-12-12
**Excellent theoretical contribution to the graph neural network literature**

**Confidence:** 4
**Recommendation:** Accept (Oral)

**Metareview:**

Graph neural networks are an increasingly popular topic of research in machine learning, and this paper does a good job of studying the representational power of some newly proposed variants. The framing of the problem in terms of the WL test, and the proposal of the GIN architecture is a valuable contribution. Through the reviews and subsequent discussion, it looks like the issues surrounding Theorem 3 have been resolved, and therefore all of the reviewers now agree that this paper should be accepted. There may be some interesting followup work based on studying depth, as pointed out by reviewer 1, but this may not be an issue in GIN and is regardless a topic for future research.